# Deep mutational scanning of essential bacterial proteins can guide antibiotic development

Liselot Dewachter [1,2] ✉, Aaron N. Brooks [3], Katherine Noon[3], Charlotte Cialek[3], Alia Clark-ElSayed[3], Thomas Schalck[1,2], Nandini Krishnamurthy [3], Wim Versées [4,5,7], Wim Vranken [4,5,6,7] & Jan Michiels [1,2,7] ✉

Deep mutational scanning is a powerful approach to investigate a wide variety of research questions including protein function and stability. Here, we perform deep mutational scanning on three essential *E. coli* proteins (FabZ, LpxC and MurA) involved in cell envelope synthesis using high-throughput CRISPR genome editing, and study the effect of the mutations in their original genomic context. We use more than 17,000 variants of the proteins to interrogate protein function and the importance of individual amino acids in supporting viability. Additionally, we exploit these libraries to study resistance development against antimicrobial compounds that target the selected proteins. Among the three proteins studied, MurA seems to be the superior antimicrobial target due to its low mutational flexibility, which decreases the chance of acquiring resistance-conferring mutations that simultaneously preserve MurA function. Additionally, we rank anti-LpxC lead compounds for further development, guided by the number of resistance-conferring mutations against each compound. Our results show that deep mutational scanning studies can be used to guide drug development, which we hope will contribute towards the development of novel antimicrobial therapies.

Deep mutational scanning is a powerful way to study protein function[1–3], stability[4], amino acid roles[5], evolvability[6], epistasis[7] and more. The power of deep mutational scanning approaches relies on the construction of large mutant libraries that contain a wide variety of gene variants, followed by the selection, evaluation and identification of these variants[8,9]. Currently, such large mutant libraries are mostly constructed through error-prone PCR[3,6], or with degenerate oligonucleotides that can be used as primers[10–12] or tiles for ORF construction[13]. While these approaches have proven successful at generating valuable biological insights, they are

limited in that library construction occurs in vitro and generates mutant alleles that are mostly studied outside of their natural context. Because expression level, copy number and genomic context can influence phenotypes, it would be advantageous to introduce mutations directly into the genome of interest rather than in vitro. High-throughput CRISPR-based editing can be used for this purpose[1]. Indeed, one of the main advantages of CRISPR-based genome editing is its scalability; targeting different DNA sequences can be done in parallel by providing different cells with different sgRNAs[14–16]. Recently, efforts in increasing the

[1]Centre of Microbial and Plant Genetics, KU Leuven, Leuven, Belgium. [2]VIB-KU Leuven Center for Microbiology, Leuven, Belgium. [3]Inscripta, Inc, Boulder, CO 80301, USA. [4]Structural Biology Brussels, Vrije Universiteit Brussel, Brussels, Belgium. [5]VIB-VUB Center for Structural Biology, Brussels, Belgium. [6]Inter-university Institute of Bioinformatics in Brussels, ULB-VUB, Brussels, Belgium. [7]These authors contributed equally: Wim Versées, Wim Vranken, Jan Michiels. ✉e-mail: liselot.dewachter@kuleuven.be; jan.michiels@kuleuven.be

throughput of CRISPR-based genome editing have led to the development of dedicated workflows that allow for the simultaneous construction of thousands of targeted genomic edits in large pooled mutant libraries that can be used for deep mutational scanning experiments[17,18].

Using automated high-throughput CRISPR-based editing of the *Escherichia coli* genome, we here perform deep mutational scanning to create full-length saturation editing libraries of three different *E. coli* proteins: FabZ, LpxC and MurA. These proteins are all essential for *E. coli* viability and are involved in the synthesis of different layers of the cell envelope. FabZ is a dehydratase involved in the synthesis of fatty acids that are used for the construction of phospholipids[19]. LpxC is needed for the production of lipid A, the lipid portion of lipopolysaccharide (LPS) which is an essential component of the outer membrane of Gram-negative bacteria[20]. Finally, the MurA enzyme catalyzes the first step in the production of peptidoglycan precursors that are used to build the rigid cell wall that helps maintain cell shape and integrity[21]. Importantly, all these proteins are considered attractive targets for the development of novel antibiotics[22–25]. Since we are currently on the verge of a world-wide health crisis due to the relentless increase in antibiotic resistance, the development of new antimicrobials and exploration of novel antibiotic targets is urgently needed[26–31]. We hope to contribute towards this goal by providing detailed functional insights into the potential drug targets FabZ, LpxC and MurA. Moreover, we use our full-length saturation editing libraries to estimate the likeliness of resistance development against lead compounds, thereby prioritizing both targets and compounds for further drug development.

## Results

### Saturation editing of *fabZ*, *lpxC* and *murA* using high-throughput CRISPR-Cas genome editing

Three essential genes that are involved in the synthesis of the Gram-negative cell envelope; *fabZ*, *lpxC* and *murA*, were chosen for full-length saturation editing. Saturation editing libraries of these genes were created using the Onyx® Digital Genome Engineering platform. Onyx® automates all the steps of genome-scale strain engineering and has a performance optimized version of CREATE technology at its core[18]. This automated platform allows for high-throughput CRISPR-based editing of the *E. coli* genome using the MAD7 CRISPR nuclease (https://www.inscripta.com/technology/madzymes-nucleases), provided on an inducible plasmid. Both the sgRNA and the repair template are provided on a second plasmid carrying a constitutive promoter for sgRNA expression in addition to a barcode to track the plasmids[18,32]. Repair templates contain the desired genomic edit and display homology to the targeted genomic site so that, upon cutting by the MAD7 enzyme, this oligo – together with the desired mutation – is incorporated into the *E. coli* genome (Fig. 1a). Apart from the desired edit, the repair template also contains one or more synonymous mutations that prevent re-cutting by eliminating the PAM site[18]. Repair templates were designed so that, at the protein level, every amino acid would be replaced by every other amino acid. Additionally, every codon was also mutated to a synonymous codon. This way, every amino acid should be targeted 20 times (19 amino acid substitutions and one synonymous change), except for methionine and tryptophan residues, for which no synonymous codons exist. No edits were designed to target the start codons of the different genes. In total, 17,415 edits (20*(150 FabZ residues + 304 LpxC residues + 418 MurA

**a** *E. coli*

sgRNA
Repair template
MAD7
Genome

sgRNA
MAD7
Genome
X          X
Repair template
Desired mutation     synonymous PAM mutation

**b**

FabZ   20*(151 residues - M1) - 5 M/W residues = 2995 mutants

```
0   MTTNTHTLQIEEILELLPHRFPFLLVDRVLDFEEGRFLRAVKNVSVNEPF 50
    FQGHFPGKPIFPGVLILEAMAQATGILAFKSVGKLEPGELYYFAGIDEAR 100
    FKRPVVPGDQMIMEVTFEKTRRGLTRFKGVALVDGKVVCEATMMCARSRE 150
    A 151
```

LpxC   20*(305 residues - M1) - 9 M/W residues = 6071 mutants

```
0   MIKQRTLKRIVQATGVGLHTGKKVTLTLRPAPANTGVIYRRTDLNPPVDF 50
    PADAKSVRDTMLCTCLVNEHDVRISTVEHLNAALAGLGIDNIVIEVNAPE 100
    IPIMDGSAAPFVYLLLDAGIDELNCAKKFVRIKETVRVEDGDKWAEFKPY 150
    NGFSLDFTIDFNHPAIDSSNQRYAMNFSADAFMRQISRARTFGFMRDIEY 200
    LQSRGLCLGGSFDCAIVVDDYRVLNEDGLRFEDEFVRHKMLDAIGDLFMC 250
    GHNIIGAFTAYKSGHALNNKLLQAVLAKQEAWEYVTFQDDAELPLAFKAP 300
    SAVLA 305
```

MurA   20*(419 residues - M1) - 11 M/W residues = 8349 mutants

```
0   MDKFRVQGPTKLQGEVTISGAKNAALPILFAALLAEEPVEIQNVPKLKDV 50
    DTSMKLLSQLGAKVERNGSVHIDARDVNVFCAPYDLVKTMRASIWALGPL 100
    VARFGQGQVSLPGGCTIGARPVDLHISGLEQLGATIKLEEGYVKASVDGR 150
    LKGAHIVMDKVSVGATVTIMCAATLAEGTTIIENAAREPEIVDTANFLIT 200
    LGAKISGQGTDRIVIEGVERLGGGVYRVLPDRIETGTFLVAAAISRGKII 250
    CRNAQPDTLDAVLAKLRDAGADIEVGEDWISLDMHGKRPKAVNVRTAPHP 300
    AFPTDMQAQFTLLNLVAEGTGFITETVFENRFMHVPELSRMGAHAEIESN 350
    TVICHGVEKLSGAQVMATDLRASASLVLAGCIAEGTTVVDRIYHIDRGYE 400
    RIEDKLRLALGANIERVKGE 419
```

= 17,415 mutants in total

**Fig. 1 | Construction of saturation editing libraries of *E. coli* FabZ, LpxC and MurA using high-throughput CRISPR genome editing. a** For CRISPR-based editing, two plasmids were introduced into individual *E. coli* cells. The first plasmid, the 'engine plasmid' encodes the MAD7 enzyme used for genomic cutting. The second plasmid, the 'barcode plasmid', encodes the sgRNA and repair template. The repair template is incorporated into the *E. coli* genome by homologous recombination and contains the desired edit as well as one or more synonymous mutations that prevent re-cutting by eliminating the PAM site. **b** In the FabZ, LpxC and MurA saturation editing libraries, every amino acid was replaced by all 19 other amino acids, except for the start codon which was not mutated. Additionally, as a control, every codon was mutated to a synonymous codon, except for methionine (M) and tryptophan (W) residues for which no synonymous codons exist. This results in a total of 20 (or 19 for M and W residues) mutations per amino acids, leading to 17,415 variants across all three libraries.

residues) – 25 methionine (M) and tryptophan (W) residues, Fig. 1b) were designed.

After library synthesis and outgrowth of the engineered bacteria, Illumina sequencing was performed to check which of the designed edits could be detected in the *E. coli* genome. Since all three proteins (FabZ, LpxC and MurA) targeted by saturation editing are essential for *E. coli* viability, protein activity can be directly evaluated by checking the presence – and therefore viability – of variants in the constructed libraries. Even though the generated libraries are barcoded[18], we directly sequenced the targeted genes in the chromosome to identify which mutations were present in the pooled mutant library. As a result, cells that received a barcoded plasmid but in which editing did not proceed correctly were not taken into account.

To interrogate the reproducibility of our high-throughput gen-ome-editing approach, two replicate Onyx® libraries were built on independent *E. coli* colonies, showing high correlation between repli-cates (Spearman's ρ for FabZ 0.946, LpxC 0.900 and MurA 0.872) (Supplementary Fig. S1). These results confirm that the deep muta-tional scanning method used here is highly reproducible. Read counts associated with the designed edits are listed in Supplementary Data 1 and 2. Further analyses were performed on the first replicate of each library.

## Editing libraries for FabZ, LpxC and MurA are almost fully saturated

We first verified the quality of the generated libraries by estimating the saturation level. Because all three targeted genes are essential, it is to be expected that some edits would not be detected even if they were successfully introduced due to drop-out of non-viable variants. Therefore, instead of looking at all designed edits to determine saturation levels, we focused on the synonymous mutations that – in principle – should have minor effects on cell viability. However, we do note that synonymous mutations are not necessarily neutral and sev-eral studies have indeed demonstrated that they can have considerable fitness effects[33–36]. It therefore remains plausible that some of the missing synonymous mutations are absent from the libraries due to detrimental effects on fitness. In this case, the saturation levels here would underestimate the true saturation levels. Of all designed synonymous edits, 96.6% were detected for FabZ, 97.3% for LpxC and 96.3% for MurA. Given this estimated saturation level of around 96% and assuming that it is similar for non-synonymous mutations, the likelihood of any specific residue not being mutated at all by random chance would be in the order of $10^{-28}$ $(=(4/100)^{20})$. Therefore, the absence of a large number of edits at a specific position could point to either biological or technical difficulties in mutating this residue.

To rule out the possibility that technical difficulties, such as inefficient PAM sites, prevent some residues from being mutated, we looked for residues that were not mutated at all, i.e. residues for which none of the 20 designs (including the synonymous design) were detected. In the first replicate of our library, we identified one such uneditable residue, MurA R120. However, the synonymous R120R edit could be detected in the second replicate of the MurA library, indi-cating that also this position can be mutated. We hypothesize that, because of the reported important role of MurA R120 in substrate binding[37–40], many edits at this position did not support viability and that any remaining mutations (such as the synonymous edit) were not detected due to random chance. Taken together, these data show that the absence of many mutations at a specific position can be used to pinpoint residues that are important for protein function.

## Competition within saturation editing libraries reveals the fit-ness effect of each edit

Libraries that were sequenced directly after construction on the Onyx® Digital Genome Engineering platform were depleted for many edits that, based on literature, are expected to interfere with protein function (Supplementary Data 1 and 2, Supplementary Fig. S2A–C). These results indicate that already at this stage the libraries contain valuable information. However, to check whether additional genera-tions of growth and competition between variants would lead to fur-ther depletion of edits that interfere with protein functionality, we performed additional selection experiments. First, libraries were grown overnight to increase starting cell numbers. They were then diluted in triplicate and grown for 5 or 15 generations. At these time points, as well as at the start of the experiment, library composition was determined by direct edit detection on the genome using Illumina sequencing (Supplementary Data 3). For each library, edits can be found for which the abundance remains unchanged, increases or decreases (Supplementary Fig. S2D–F). However, not all libraries behave the same. Selection of the FabZ library has a stronger effect on library composition and leads to depletion of a larger number of edits than is the case for LpxC or MurA (Fig. 2). It therefore appears to take longer before detrimental FabZ edits affect growth and viability, compared to LpxC or MurA. This is in line with a recent CRISPRi depletion study that showed that growth is more robust against depletion of FabZ than LpxC and MurA[41], indicating that in comparison to LpxC or MurA, there is an excess of FabZ activity present in the cell.

Analysis of the changes in abundance of all variants across the selection experiment also allowed estimation of the fitness effect associated with each edit. To do so, the change in abundance of each mutation was fitted with a weighted least squares log-linear model using Enrich2[42]. The slope of the fit was taken to be a mutation's competition coefficient and was normalized to total read counts and adjusted so that the most common competition coefficient was set to zero (Supplementary Data 4). Competition coefficient values greater than zero indicate improved growth, whereas coefficients less than zero indicate impaired growth. Using this approach, each edit was associated with a competition coefficient indicative of its fitness effect (Supplementary Fig. S3). We observed hundreds of mutations that, while tolerated, confer a large fitness burden (*fabZ*: 107 [4.7%]; *lpxC* 161 [3.29%]; *murA* 164 [2.79%]; gene #mutations [% total mutations]), where significant fitness burden is defined as mutations with a competition coeffi-cient ≤two standard deviations from the most common coefficient, zero (Supplementary Fig. S4A). Additionally, more extreme com-petition coefficients (typically negative) were observed at amino acid positions with higher dropout rates (Supplementary Fig. S4B).

## Saturation editing libraries identify residues that are important for protein function

Because the selection experiment revealed the continued dropout of non-viable or severely defective mutations from the FabZ library, we decided to focus further analyses of important residues on the selec-ted libraries that were grown for 15 additional generations. At this time point, the dropout fraction and composition of all libraries largely stabilizes (Fig. 2 and Supplementary Data 3). The presence or absence of edits in these libraries therefore provides a strong indication for whether a specific amino acid substitution can support protein func-tion and viability or not, although we note that some viable edits might have been outcompeted at this stage if they were associated with strong fitness defects.

Although not all amino acid changes are allowed, the number of tolerated amino acid substitutions (i.e. substitution that could be detected in the library after 15 generations of growth) is surprisingly high for most positions (Fig. 3a–c): around 50% of all residues could be mutated to all or all but one amino acid(s) (Fig. 3d). Our results thereby highlight the robustness of protein function in light of single amino acid changes. Interestingly, tolerance for amino acid changes is protein dependent. Of the three tested proteins, MurA is the least tolerant, i.e. more designed edits were lost from the library because they were unable to support viability.

 

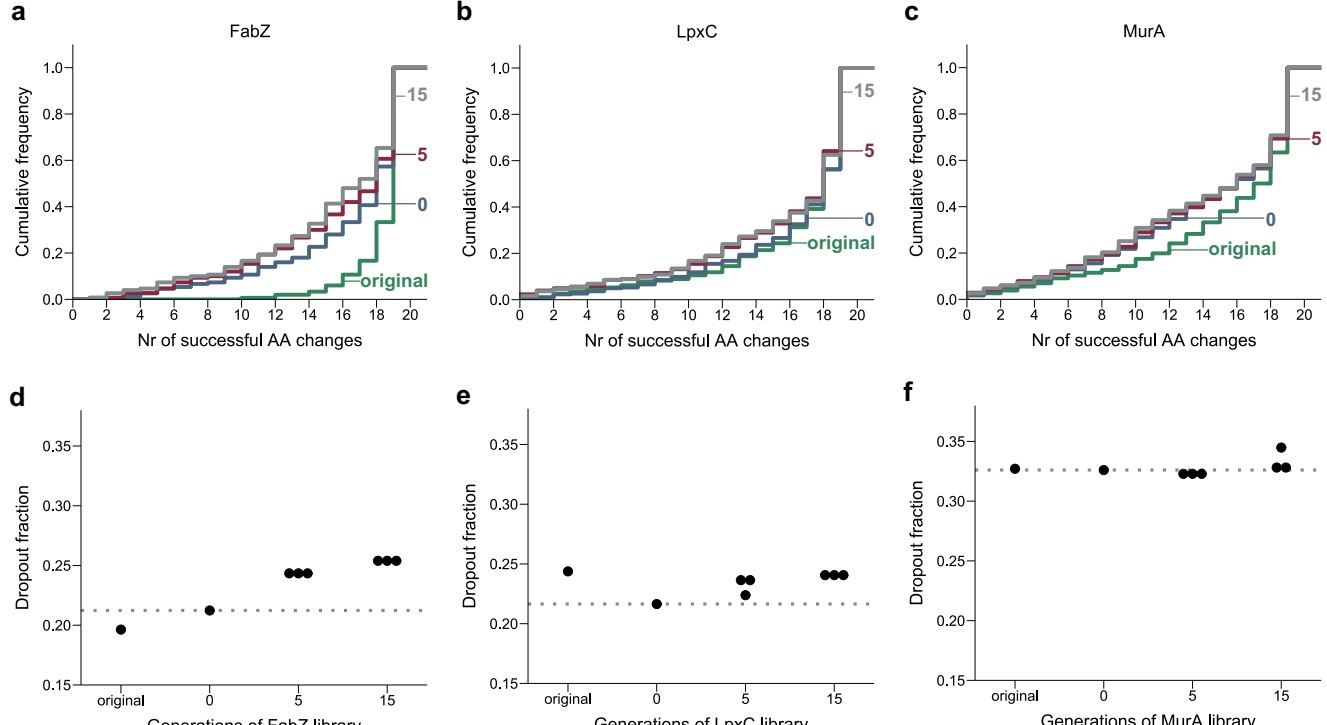

**Fig. 2 | The stability of library composition across growth cycles varies for different proteins.** The cumulative frequency distribution of successful amino acid substitutions per residue, i.e. edits that could be detected by sequencing, is shown for the FabZ (**a**), LpxC (**b**) and MurA (**c**) libraries. Different curves represent the library composition at different points of sequencing. In case multiple replicates were sequenced (for samples 5 and 15), the first replicate is shown. This figure shows the dropout fraction, i.e. the fraction of edits detected by fewer than 10 reads, in function of the number of generations the FabZ (**d**), LpxC (**e**) or MurA (**f**) libraries were grown. As a reference, the dropout fraction at the start of the selection experiment is indicated by a dotted line. Original refers to the libraries immediately after construction. Generations 0, 5 and 15 refer to the number of generations libraries were grown as part of the selection experiment. As expected, cycles of growth will lead to an increased loss of amino acid substitutions, resulting in flatter cumulative distribution curves and larger dropout fractions. Source data are provided as Supplementary Data 3. AA amino acid.

Additionally, plotting the presence or absence of each edit in the sequenced libraries, as well as their change in abundance throughout 15 generations of growth, provides an indication of which residues are important for protein function and what type of substitutions are allowed and therefore do not completely abolish functionality and viability (Fig. 4).

In order to further pinpoint positions that are important for protein function, we assigned a tolerance score to each residue based on the number and types of amino acid substitutions that are tolerated, i.e. that are detected in the saturation editing libraries after selection. Although complex interpretations exist for assessing amino acid similarities[5], we here use a simple normalized amino acid similarity score[43], where fully tolerant promiscuous sites obtain a score of 1, fully intolerant ones a score of 0. Tolerated mutations to very biochemically and/or structurally different amino acids (e.g. Gly to Trp) receive higher scores than ones between similar amino acids (e.g. Leu to Ile). A site with few tolerant mutations between very different amino acids might therefore receive a higher tolerance score than a site with more mutations between similar amino acids. Tolerance scores are listed in Supplementary Data 5 and the distribution of these scores is shown in Fig. 5a–c.

Residues with low tolerance scores could be important for protein function due to several different reasons. For example, they could be part of the catalytic site, be involved in protein-protein interactions or influence protein folding and stability. To distinguish between some of these options, we calculated the Relative Solvent Accessibility (RSA) of individual residues, which is a measure for how exposed an amino acid is to the cellular environment. Residues with low RSA values are buried inside the protein and are therefore unlikely to play a direct role in

protein function, but rather contribute to folding and/or stability. RSA values were extracted from relevant protein structures in the PDB and are displayed in Fig. 5d–f and listed in Supplementary Data 5. For each protein, around 10 residues that might play a direct role in protein function were selected for further evaluation (Fig. 5d–f, Table 1). These are (partially) exposed residues (RSA > 1%) that display the lowest tolerance scores for their respective libraries. Several of the selected residues were previously shown to be important for protein function. However, we also identify residues that were not yet implicated in protein function, thereby expanding our insight into these essential bacterial proteins. All selected residues together with their previously reported functions are listed in Table 1.

For LpxC, our analysis revealed three positions that can only be substituted by a synonymous codon: H79, D242 and D246. H79 and D242 are required for coordinating the catalytically important $Zn^{2+}$ ion of LpxC, together with H238 that is seemingly more tolerant for substitutions (7 substitutions allowed)[44]. D246 directly interacts with the presumed catalytic residue H265, and was previously suggested to affect the orientation and/or charge of H265[45]. Indeed, the latter residue was proposed to act as the general acid required to protonate the amino leaving group in the LpxC-catalyzed deacetylase reaction[45–48]. Interestingly, we find that substitution of H265 by a glutamine residue is tolerated, while the latter can clearly not act as a general acid. Similarly, also the substitution of E78, the presumed general base that deprotonates the $Zn^{2+}$-bound nucleophilic water molecule[49], by a valine residue seems to sustain viability. Such unexpected tolerated mutations indicate the complexity of protein functionality within the in vivo cell context, as opposed to in vitro experiments. In addition to these residues, several other residues with low tolerance scores and

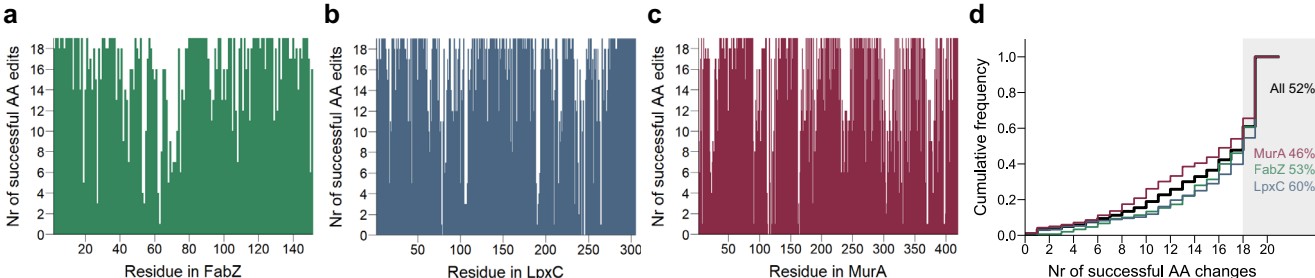

**Fig. 3 | Saturation editing libraries reveal a surprisingly high tolerance for amino acid substitutions in three essential *E. coli* proteins.** Bar plots show the number of successful amino acid changes at each position of the proteins FabZ (**a**), LpxC (**b**) and MurA (**c**). Successful amino acid changes were defined as edits that were found to be present in the library after 15 generations of growth (average read count >0). **d** The cumulative frequency distribution of successful amino acid substitutions detected after 15 generations of growth is shown for all libraries taken together (All) or the FabZ, LpxC and MurA library separately. The percentage of residues that tolerates all or all but one amino acid changes (≥18) is stated and highlighted at the right side of the figure. Source data are provided as Supplementary Data 3. AA amino acid.

**a**

**b**

**c**

**Fig. 4 | Analysis of saturation editing libraries can be used to identify important residues.** These heat maps display the presence or absence of each amino acid substitution in the saturation editing library of FabZ (**a**), LpxC (**b**) and MurA (**c**). Edits that were not detected after 15 generations of growth are shown in white (mean read count = 0). The mean normalized read counts of other edits are shown on a blue color scale. To obtain these values, read counts after 15 generations of growth were normalized to the read counts of the same edit at the start of selection (0 generations). The average of these normalized read counts of all three repeats was taken as input for these heat maps. Those edits that had a read count of zero at the start of selection, but a non-zero read count after 15 generations of growth are shown in yellow. Source data are provided as Supplementary Data 3.

Read counts at 15 generations, normalized to read counts at 0 generations

≥ 1
0.75
0.50
0.25
0

Read count 0 at 0 generations and > 0 at 15 generations

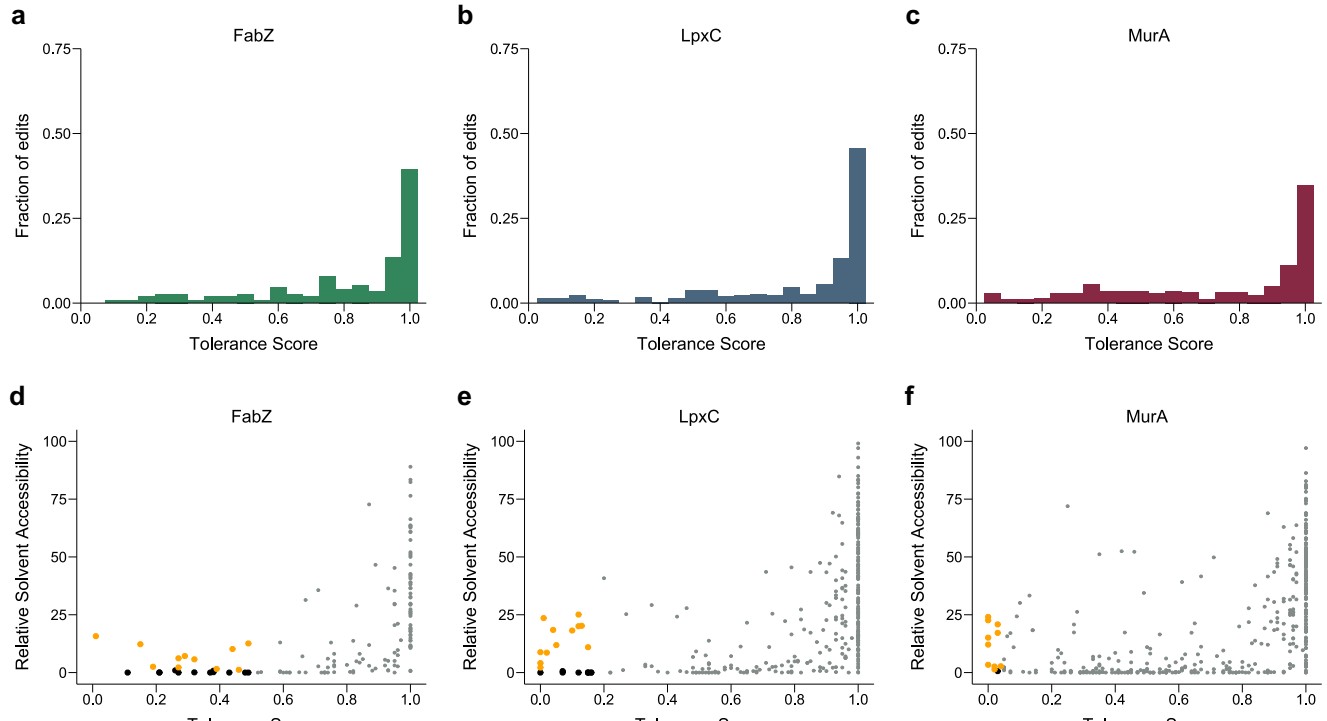

**Fig. 5 | Analysis of the number and types of amino acid substitutions present at each position of the saturation editing libraries can be used to identify important residues.** The distribution of tolerance scores, i.e. how tolerant each residue is towards substitutions with different amino acids, is shown for FabZ (**a**), LpxC (**b**) and MurA (**c**). The relative solvent accessibility (RSA) is plotted in function of the tolerance score for each residue of FabZ (**d**), LpxC

(**e**) and MurA (**f**). Residues with the lowest tolerance scores and low RSA are colored black and are likely essential for protein folding and stability. The residues with the lowest tolerance scores and relatively high RSA are highlighted in orange and likely play an important and direct role in protein function. Source data are provided as Supplementary Data 5.

high RSA values - which thus likely directly participate in protein function - were identified in LpxC, as listed in Table 1. Not entirely unexpected, most of these residues are located within or around the active site/substrate binding pocket. G264 is located in close spatial proximity to H265 and to the pyrophosphate groups of the UDP moiety of the substrate, and substitution with bulky amino acids would likely interfere with substrate binding. Another set of residues that display a limited tolerance to mutations are either located within (R190, T191, F192) or interact with (D105) the R190-G193 region that directly interacts with the glucosamine moiety of the substrate and contributes to catalysis[50]. Also K239 makes a direct hydrogen bond with the substrates' glucosamine moiety, and a K239A mutation was previously reported to affect catalysis[50]. In this respect it is remarkable that mutations to Met and Asn are allowed, while no other mutations are identified in our analysis. A final functional category of residues with low mutational tolerance consists of residues that line the acyl-binding groove (G210 and A215). Mutation of these small residues to amino acids with larger side chains would likely affect substrate binding.

A similar analysis on MurA reveals several residues that cannot be replaced by any other amino acid. These include C115, the proposed general acid required to protonate the C3 atom of the phosphoenolpyruvate (PEP) substrate in the MurA-catalyzed reaction[51], as well as G114, G118, G164 and D231. It is remarkable that we find C115 to be absolutely essential in both replicates of our library, while it was previously reported that a C115D mutation retains catalytic activity[51]. In this context it is worth mentioning that we also retrieve K22 as a residue with relatively little tolerance to mutations (tolerance for F, I, N, T). Eschenburg et al. proposed this residue to be the general acid that protonates PEP[39]. However, the latter proposal does not agree with the K22F, K22I, K22N and K22T mutations that we retrieve as

being viable. We also identify residues that were until now not described to play an essential role in MurA, including G114, G118 and D231. G114 and G118 are located in the P112-P121 catalytic loop harboring the C115 general acid, and are most likely crucial to maintain the loop conformation and to sustain the required conformational changes within this loop[51]. Within the same P112-P121 loop we also identified R120, which plays a previously described role in substrate binding[37–40]. Likewise, also the essential G164 residue interacts via its main chain NH group with the pyrophosphate group of the UDP moiety of the substrate. The essential nature of D231 is rather unexpected as this residue is located at relatively large distance from the substrate. However, closer inspection shows that this residue interacts with the main chain NH group of K22, and might be required to properly orient K22 to exert its function. Moreover, the D231-K22 interaction is located at the interface of the two MurA domains and might thereby contribute to the integrity of the proteins' tertiary structure, as previously noted[52]. A number of other residues can only be substituted by a single other amino acid. The side chain of S162 makes a direct hydrogen bond with the pyrophosphate group of the substrate, which explains its tolerance for substitution with a threonine only. R331 in turn was previously already suggested to interact with the substrate PEP, and, correspondingly, can only be replaced with a functionally very similar lysine residue[39,40]. D369, which can only be replaced with a functionally similar glutamate residue, is located further away from the substrate but is localized in between the important C115 and R331 residues. G398 is located adjacent to R397, which was proposed previously to play an important role in the product release mechanism[51] and can only be substituted to a serine residue in our study. Finally, D305 was described to be essential for catalysis[53,54], and a role as general base required for deprotonation of the C3 hydroxyl of the UDP-GlcNAc substrate was proposed[38]. In theory the observed (viable) D305E, D305H and D305Y

**Table 1 | Selected surface exposed residues with relatively low tolerance scores for FabZ, LpxC and MurA**

| Protein | Residue | Tolerance score | Allowed substitutions | Previously described function |
|---|---|---|---|---|
| FabZ | H19 | 0.27 | H,L,M,N,Q,T | |
| FabZ | H54 | 0.15 | H,Q,V,Y | Catalytic residue[25,55,56,75] |
| FabZ | F55 | 0.44 | F,G,I,K,L,M,N,Q,V,W,Y | |
| FabZ | P62 | 0.19 | A,M,P,V,W | |
| FabZ | G63 | 0.01 | A,G | |
| FabZ | E68 | 0.29 | A,D,E,F,L,V | Catalytic residue[25,56,75] |
| FabZ | A71 | 0.32 | A,C,E,L,N,S,T,V | |
| FabZ | Q72 | 0.39 | A,E,G,H,L,N,Q,S | |
| FabZ | F93 | 0.49 | C,F,G,I,L,M,N,Q,S,V,W,Y | |
| FabZ | G108 | 0.27 | A,C,G,P,Q,S,T,W | |
| FabZ | G129 | 0.46 | A,C,D,F,G,L,N,P,S,T,V,W | |
| LpxC | H79 | 0.00 | H | Catalytic $Zn^{2+}$ coordination[45,48,76] |
| LpxC | D105 | 0.13 | C,D,E,G | Protein stability[48] |
| LpxC | R190 | 0.02 | K,R | |
| LpxC | T191 | 0.12 | C,P,S,T | Substrate binding[45,76], stabilizing intermediates[47,50] |
| LpxC | F192 | 0.12 | F,I,M,W,Y | Substrate binding[45,46,50,76] |
| LpxC | G210 | 0.01 | A,G | Substrate binding[45] |
| LpxC | A215 | 0.15 | A,C,G,S,T,V | Substrate binding[45,77] |
| LpxC | K239 | 0.10 | M,N | Substrate binding[46,50,76,78,79] |
| LpxC | D242 | 0.00 | D | Catalytic $Zn^{2+}$ coordination[45,76], substrate binding[46] |
| LpxC | D246 | 0.00 | D | Catalytic residue[45,47,48] |
| LpxC | G264 | 0.04 | G,S | |
| LpxC | H265 | 0.05 | H,Q | Catalytic residue[45–48] |
| MurA | G114 | 0.00 | G | |
| MurA | C115 | 0.00 | C | Catalytic residue[38,66,80,81] |
| MurA | G118 | 0.00 | G | |
| MurA | R120 | 0.03 | H,R | Substrate binding[37–40] |
| MurA | S162 | 0.03 | S,T | Substrate binding[37,38] |
| MurA | G164 | 0.00 | G | Substrate binding[37,38] |
| MurA | D231 | 0.00 | D | |
| MurA | R331 | 0.02 | K,R | Substrate binding[37,39,40] |
| MurA | D369 | 0.02 | D,E | |
| MurA | G398 | 0.04 | G,T | |

For each library, approximately 10 residues with the lowest tolerance scores of that library were chosen. The substitutions that are encountered at these positions in the saturation editing libraries are listed as allowed substitutions. Source data are provided as Supplementary Data 5.

variants could take over such a role, although the available space to allow substitution of D305 with bulkier imidazole and phenol groups seems limited.

The mutational analysis of FabZ presents a more complex and intriguing image, as all FabZ residues are quite tolerant to substitutions. This is particularly remarkable for the residues H54 and E68, which were proposed to act as the general base and general acid, respectively, in the FabZ-catalyzed dehydration of the β-hydroxyacyl-ACP[55,56]. Our observation that H54 can be substituted to three other amino acids (Q, V, Y), and that E68 can be substituted to five other amino acids (including the non-polar residues A, L, V, F), seems to contradict with an essential function for these residues. Several other (partially) surface exposed residues (RSA > 1) also show a somewhat lower tolerance score for substitutions, as listed in Table 1. The residue least tolerant to substitutions is G63, which can only be replaced with alanine, and which borders the surface of the acyl-binding tunnel. Most probably substitution by a bulkier amino acid would sterically interfere with substrate binding. Additionally, the main chain NH group of G63 is within hydrogen bond distance to the carbonyl group of the substrate's β-hydroxyacyl moiety. Many other residues that turn up with a lower tolerance score are located within the acyl-binding tunnel,

including: H19, F55, P62, A71, Q72 and F93. Finally, G108 is located at the subunit interface of the FabZ hexameric trimer of dimer arrangement. Despite its lower tolerance score, we were surprised to observe that certain substitutions with bulky amino acids (e.g. Q, W) are still allowed. Mutations at a single position might not be sufficiently disruptive to interfere with multimer formation, thereby highlighting a fundamental restriction of this approach, which is limited to single isolated mutations and cannot investigate co-occurring mutations that might be synergistic or compensatory.

Taken together, our results demonstrate that saturation editing of essential genes combined with the identification of viable amino acid substitutions can be used to pinpoint important residues and can reveal novel insights into protein function.

**Saturation editing libraries can guide efforts for the development of novel antibiotics**

Finally, we aimed to exploit the saturation editing libraries of essential *E. coli* proteins FabZ, LpxC and MurA to formulate recommendations for antibiotic development. First, to identify surface exposed regions that are important for protein function and could be targeted by antimicrobial compounds, we plotted the tolerance scores for all

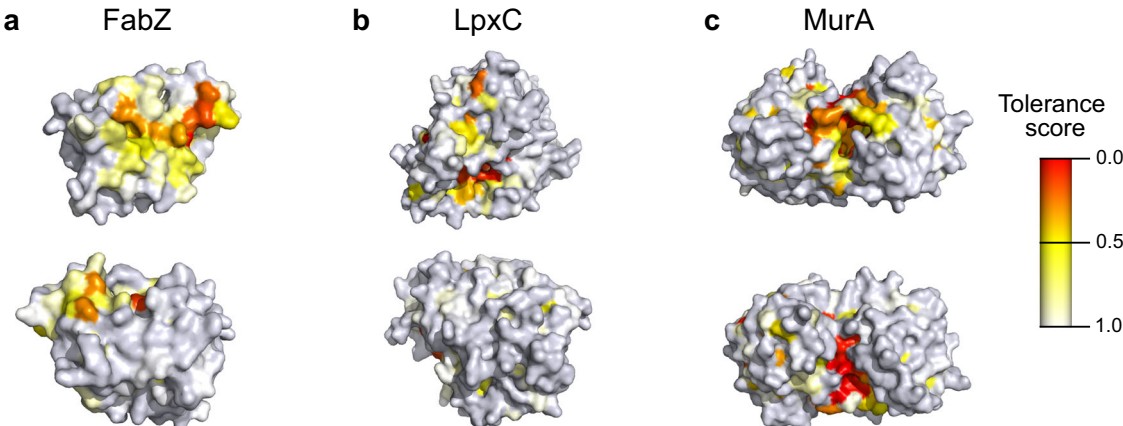

**Fig. 6 | Protein structures colored by each residue's tolerance score reveal regions essential for protein function that can be targeted by antimicrobial compounds.** Tolerance scores calculated here were plotted onto experimentally determined protein structures for FabZ, PDB 6n3p (**a**); LpxC, PDB 4mqy (**b**); and MurA, PDB 1uae (**c**). For each protein, two different surfaces are shown at the top and bottom. PDB files containing tolerance scores and corresponding PyMol scenes are provided as supplementary information. Source data are provided as Supplementary Data 5 and PyMol scenes are provided as Supplementary Data 7.

residues onto the corresponding protein structures (Fig. 6). As expected, these augmented protein structures reveal the importance of the catalytic site for protein function, but could in theory also reveal sites involved in allosteric regulation, protein-protein interactions, etcetera.

Additionally, since our saturation editing libraries provide information on the protein's mutational tolerance, they can be used to make predictions regarding resistance development. This way, drug development efforts can be guided towards compounds and targets that are the least susceptible to acquiring resistance mutations. As is clear from Figs. 3 and 4, the mutational flexibility of FabZ, LpxC and MurA differs. Whereas the saturation levels for these libraries are almost identical (96–97%), the percentage of mutations that is tolerated varies, with 84% of mutations tolerated for FabZ (2506 detected mutations out of 2995 designed mutations), 84% for LpxC (5105 detected mutations out of 6071 designed mutations) and 76% for MurA (6352 detected mutations out of 8349 designed mutations). The same trend emerges when calculating the percentage of residues that tolerates all or all but one amino acid changes. This number reaches 53% for FabZ, 60% for LpxC and 46% for MurA (Fig. 3D). When looking only at surface-exposed residues (RSA > 1), i.e. residues relevant for antibiotic resistance development, the same trend emerges. For FabZ, LpxC and MurA, respectively 89, 88 and 82% of all edits that target surface-exposed residues are tolerated. This corresponds to 68, 70 and 61% of surface-exposed residues that tolerate all or all but one amino acid changes for FabZ, LpxC and MurA respectively. Taken together, these data indicate that, even though all three proteins are essential for *E. coli* viability, their tolerance to amino acid changes differs, with MurA being the least tolerant. Based on these data, we speculate that MurA is the least likely to develop resistance-conferring mutations when serving as an antibiotic target and is therefore the best target to pursue.

To investigate this hypothesis in more detail, we isolated library variants that are resistant to selected compounds. Fosfomycin, a known antibiotic that targets MurA directly[57], was used to select *murA* resistant variants. LpxC-targeting compounds CHIR-090[58,59] and PF-04753299 (Pfizer) were used to interrogate resistance development through *lpxC* mutations. Additionally, since it has been shown that resistance to anti-LpxC compounds can develop through mutations in *fabZ* that restore the disturbed balance between LPS and phospholipid synthesis[60–62], we also selected the FabZ library against both of these compounds. To select for resistant variants, libraries were plated onto medium containing different concentrations of the selected compounds (4x, 8x and 32x the Minimal Inhibitory Concentration (MIC),

see Methods). Colonies that were able to grow overnight were selected and their *fabZ*, *lpxC* or *murA* gene was sequenced to identify potentially resistance-conferring mutations. All isolated mutations are listed per condition in Supplementary Data 6, while a condensed form of these data is shown in Table 2.

Whereas all colonies from the FabZ or LpxC libraries selected for resistance to either CHIR-090 of PF-04753299 carry a mutation in respectively *fabZ* or *lpxC*, this is not true for selection of the MurA library against fosfomycin. In this case, the vast majority of selected resistant clones still contain a wild-type *murA* gene, pointing towards the existence of spontaneous resistance mutations that arose elsewhere in the genome. Indeed, when comparing the number of resistant variants present in the libraries to the number of spontaneous resistant variants present in a culture of the wild-type strain, they are highly similar when selecting for fosfomycin resistance (Fig. 7a–c). These data thereby confirm that fosfomycin resistance mostly arises through spontaneous resistance mutations that are not located in the *murA* gene. Nonetheless, a few *murA* variants were picked up when selecting the MurA library for fosfomycin resistance. However, each of these mutations was only found once, thereby making us question their role in mediating fosfomycin resistance. To check whether these *murA* mutations are causal to resistance or are hitchhikers present in a genome that also contains spontaneous resistance mutations elsewhere, we transferred the *murA* mutant alleles to a clean genetic background that has never been exposed to fosfomycin. Since none of these transferred mutations were able to increase MIC levels towards fosfomycin (Supplementary Table S1), we conclude that also for these selected variants, causal spontaneous mutations are located elsewhere in the genome. In conclusion, not a single *murA* mutation could be found that provides resistance to fosfomycin.

On the other hand, Fig. 7a–c shows that the number of variants resistant to CHIR-090 or PF-04753299 from either the FabZ or LpxC libraries exceeds the number of spontaneous resistant variants by several orders of magnitude, indicating that the isolated *fabZ* and *lpxC* mutations are likely causal to resistance.

Table 2 and Supplementary Data 6 show that there is a large variety in possible *fabZ* mutations that provide resistance against either CHIR-090 or PF-04753299. In fact, out of the 35 sequenced variants from the FabZ library that were either resistant against CHIR-090 or PF-04753299, 33 or 24 unique mutations were found, respectively, indicating that the search for resistant variants was not saturated and that additional resistance-conferring mutations probably

**Table 2 | Library variants that are resistant to CHIR-090, PF-04753299 or fosfomycin**

| FabZ | | | | LpxC | | | | MurA | |
|---|---|---|---|---|---|---|---|---|---|
| CHIR-090 | | PF-04753299 | | CHIR-090 | | PF-04753299 | | Fosfomycin | |
| Mutated residue | Times found | Mutated residue | Times found | Mutated residue | Times found | Mutated residue | Times found | Mutated residue | Times found |
| T3$_{E}$ | 1 | | | I2$_{K/Y}$ | 2 | I2$_{K/Y}$ | 3 | V16$_{M}$ | 1 |
| L8$_{R}$ | 1 | | | R5$_{K/N}$ | 6 | R5$_{K/N/S/V}$ | 13 | D51$_{A}$ | 1 |
| H19$_{Q}$ | 1 | | | R9$_{H}$ | 1 | | | V228$_{I}$ | 1 |
| | | F23$_{H/T}$ | 2 | | | I10$_{E}$ | 1 | R267$_{E}$ | 1 |
| | | L25$_{K}$ | 1 | V11$_{E/I/K/N/T}$ | 9 | V11$_{E/F/I/T}$ | 9 | I402$_{W}$ | 1 |
| G35$_{Y}$ | 1 | | | Q12$_{H/N}$ | 3 | | | wt | 30 |
| | | F51$_{C/P}$ | 2 | T14$_{D/E/F/S/Y}$ | 8 | T14$_{C/F/Y}$ | 7 | | |
| | | I60$_{C}$ | 1 | G15$_{H/N}$ | 2 | G15$_{H}$ | 1 | | |
| G75$_{S}$ | 1 | A71$_{G}$ | 2 | V16$_{E}$ | 1 | | | | |
| | | L90$_{C}$ | 1 | L18$_{C/V}$ | 2 | | | | |
| Y92$_{A/S/T}$ | 3 | Y92$_{Q/T}$ | 2 | A31$_{N}$ | 1 | T35$_{D}$ | 1 | | |
| F93$_{I}$ | 1 | F93$_{I}$ | 4 | wt | 0 | wt | 0 | | |
| G95$_{Q/V}$ | 2 | | | | | | | | |
| I96$_{W}$ | 1 | I96$_{W}$ | 1 | | | | | | |
| E98$_{P}$ | 1 | | | | | | | | |
| A99$_{L}$ | 1 | | | | | | | | |
| R100$_{C/S/T/Y}$ | 6 | R100$_{Y}$ | 1 | | | | | | |
| F101$_{V}$ | 1 | | | | | | | | |
| | | K102$_{L}$ | 1 | | | | | | |
| D109$_{L}$ | 1 | | | | | | | | |
| R121$_{G}$ | 1 | R121$_{F/Q}$ | 2 | | | | | | |
| R122$_{W}$ | 1 | | | | | | | | |
| L124$_{K/N}$ | 2 | | | | | | | | |
| T125$_{H}$ | 1 | | | | | | | | |
| R126$_{N/Q}$ | 2 | R126$_{G/H}$ | 4 | | | | | | |
| F127$_{Q/W}$ | 2 | | | | | | | | |
| | | G129$_{I/L}$ | 6 | | | | | | |
| V138$_{L}$ | 1 | | | | | | | | |
| | | C139$_{Y/L}$ | 2 | | | | | | |
| | | A141$_{M/Q}$ | 3 | | | | | | |
| M144$_{H/R}$ | 2 | | | | | | | | |
| A146$_{R}$ | 1 | | | | | | | | |
| wt | 0 | wt | 0 | | | | | | |

Variants were selected by plating the FabZ, LpxC or MurA library on medium with different concentrations of the indicated compounds (4x, 8x or 32x MIC). Colonies that were able to grow were sequenced to identify possible resistance-conferring mutations. This table lists the residues that were found to be mutated together with the number of times they were targeted and the detected amino acid substitutions (in subscript). For each library-compound combination, 35 resistant clones were sequenced. Source data are provided as Supplementary Data 6.
*Wt* wild type.

exist. Mapping the isolated mutations onto the FabZ protein structure demonstrates that the resistance-conferring *fabZ* mutations occur throughout the entire protein with a few preferred hotspots for mutations (Fig. 7d).

Similarly, *lpxC* also displays hotspots for resistance-conferring mutations. However, the number of different resistance-conferring mutations in LpxC is much more limited than for FabZ. Out of the 35 sequenced variants from the LpxC library that were either resistant against CHIR-090 or PF-04753299, 22 and 16 unique mutations were found, respectively. Given the relatively large number of *lpxC* mutations that were isolated multiple times, we suspect that our selection for resistant variants was more or less saturated and that most resistance-conferring *lpxC* mutations were identified. Interestingly, these mutations are exclusively found in the N-terminus of the protein

(Fig. 7e) which for CHIR-090[59], and presumably also PF-04753299, is not where the compound binds.

To estimate how likely spontaneous mutations in *fabZ*, *lpxC* or *murA* would generate resistance, we classified the number of unique resistance-conferring mutations selected here according to the minimal number of (SNPs) needed to result in the corresponding amino acid substitution (Fig. 7f–g). From these data it is clear that there are more 1 SNP mutations located in *fabZ* and *lpxC* that provide resistance against CHIR-090 than PF-04753299, meaning that resistance can likely more easily develop against CHIR-090. PF-04753299 is therefore a more attractive anti-LpxC compound. No mutations in *murA* were identified to provide resistance against fosfomycin, although we have established that spontaneous resistance mutations to this antibiotic can easily arise elsewhere in the genome.

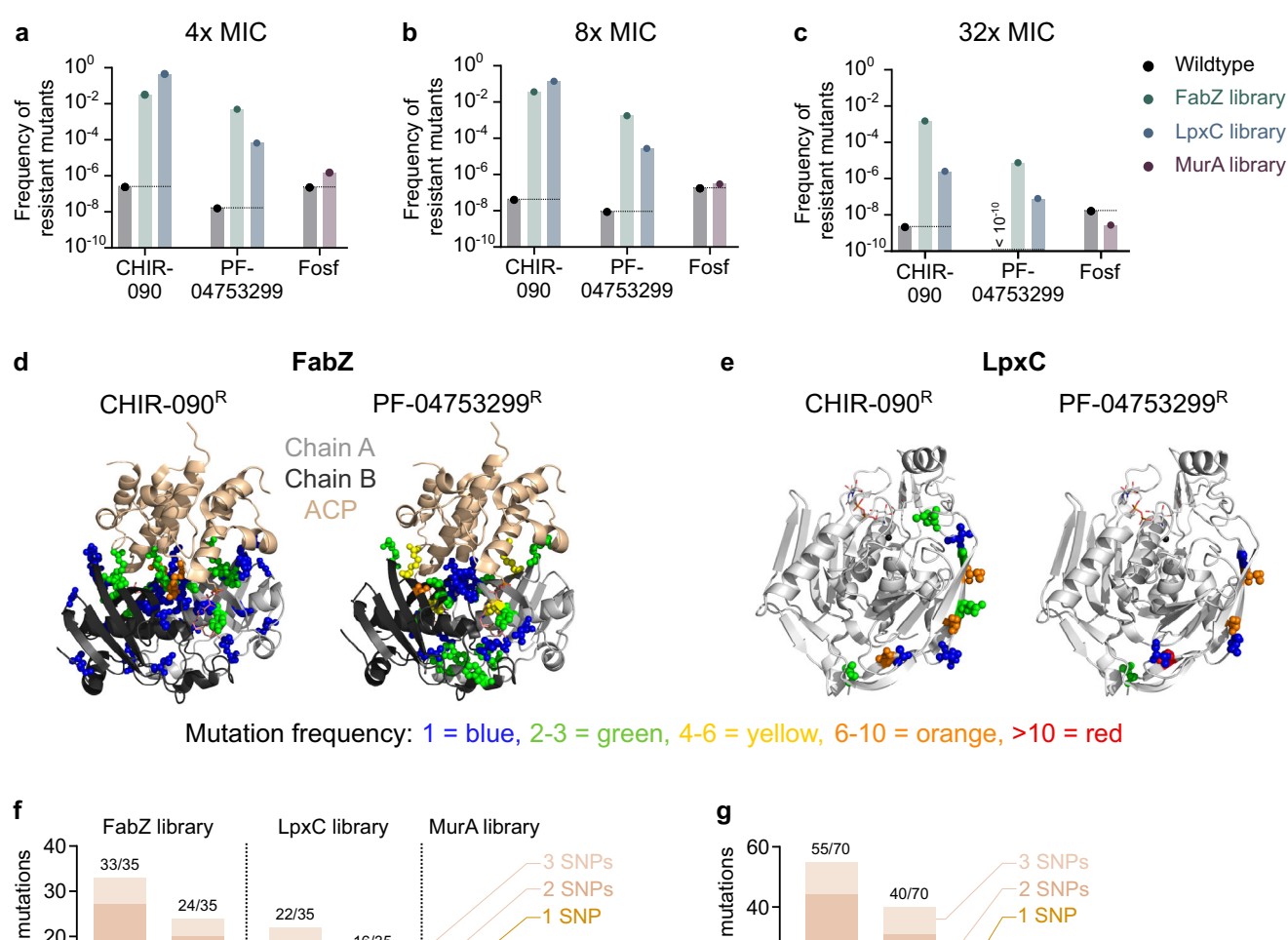

**Fig. 7 | Saturation editing libraries can guide efforts for the development of novel antibiotics. a–c** The frequency of occurrence of spontaneous resistance mutations is compared to the frequency of occurrence of resistant variants in the saturation editing libraries. This was done by plating either a wild-type culture or the different libraries onto medium containing different concentrations of the compound, i.e. 4x MIC (**a**), 8× MIC (**b**) or 32× MIC (**c**), and counting the number of colonies that developed after overnight growth. These numbers were normalized to the total cell numbers present in the wild-type culture or the libraries, respectively. **d** The location of targeted residues in the FabZ protein is shown for both CHIR-090 and PF-04753299. Residues are colored according to the number of times they were targeted in isolated resistant variants. Only one dimer of the FabZ hexamer is shown for clarity. Mutations are indicated in both chain A and B. **e** The

location of targeted residues in the LpxC protein is shown for both CHIR-090 and PF-04753299. Residues are colored according to the number of times they were targeted in isolated resistant variants. The number of unique mutations in *fabZ* and *lpxC* that provide resistance to CHIR-090 or PF-04753299 and the number of mutations in *murA* that provide resistance to fosfomycin are shown, either grouped per library and compound (**f**) or grouped per compound only (**g**). For each library-compound combination, 35 resistant variants were isolated and their *fabZ*, *lpxC* or *murA* gene was sequenced. The mutations found are subdivided into categories based on the minimal number of SNPs necessary to provide the observed amino acid change. Source data are provided as a Source Data file. Fosf fosfomycin, ACP acyl carrier protein.

## Discussion

We have performed saturation editing on three essential *E. coli* proteins using high-throughput CRISPR genome editing to identify amino acid residues that are important for protein function. We were able to confirm the role of previously annotated residues while also providing new insights into protein function and formulating recommendations for antibiotic development.

To identify essential residues, we have introduced a tolerance score that reflects how well amino acid changes are tolerated at each position in a protein. By combining this tolerance score with the relative solvent accessibility (RSA) of each residue, we made a distinction between residues that are likely important for protein folding or stability (low tolerance score and low RSA) and residues more

directly involved in protein function (low tolerance score and high RSA). This way, we identified several important or essential residues in each protein, some of which were not previously known to play an important role. Notably, our results highlight some key differences with previously obtained results from in vitro protein activity tests. For example, in MurA we find C115 to be completely intolerant to mutations in vivo, while previously a C115D mutant was shown to retain activity in vitro. These findings stress the importance of complementing any insights obtained in vitro with experiments that interrogate behavior in the much more complex in vivo setting.

Surprisingly, we found that protein function is very robust in light of mutations. The vast majority of single amino acid substitutions still support cell viability and protein function. No less than 84% of the

designed *fabZ* and *lpxC* mutations allow for viable progeny. This number is 76% for MurA. Moreover, 53, 60 and 46% of residues could be changed to all or all but one amino acid for the FabZ, LpxC and MurA proteins, respectively. Therefore, although all three proteins display a surprisingly high tolerance for amino acid changes, this tolerance level differs with MurA being the least tolerant. This difference could in part be related to the surface-to-volume ratio of different proteins and, related, the percentage of exposed residues. Our data indeed confirm that surface exposed residues can more easily tolerate mutations than buried residues[63,64]. Larger proteins that tend to have a larger surface-to-volume ratio and a higher percentage of exposed residues are therefore expected to tolerate a higher percentage of mutations than proteins that tend to have more buried residues. Indeed, MurA, the largest protein under investigation, tolerates the lowest number of mutations. Also the percentage of fully buried residues (RSA = 0), 10, 10 and 15% for FabZ, LpxC and MurA respectively, fits well with this explanation. However, differences in protein size cannot explain why a number of important FabZ residues, e.g. the catalytic residues H54 and E68, can still be replaced by several other amino acids. On the one hand, it could be that even after all performed growth steps, non-viable edits are still not depleted from the library. The generally low competition coefficients at these positions appear to suggest that, at least for some of them, this might be the case. On the other hand, low levels of mistranslation[65] that restore the mutant protein to its original form could potentially rescue cells if the amount of active protein needed is rather low. Whereas FabZ is essential for *E. coli* viability, a recent CRISPRi depletion study indeed indicated that this enzyme is present in excess compared to the other proteins investigated here[41]. Whatever the cause, the difference in tolerance to mutations between these three potential drug targets also has implications for drug development. Because MurA is the least tolerant to mutations and contains the highest number of residues that can only be replaced by a few very specific amino acids, MurA appears to be the most promising drug target.

Finally, we have tested this prediction by probing resistance development against existing antibacterial compounds through target modification. Based on the selection for *murA* variants that are resistant to the MurA-targeting antibiotic fosfomycin[57], we confirm that MurA is indeed a highly attractive antibiotic target. Even though several resistant variants could be isolated from the MurA saturation editing library, none of them carried a *murA* mutation that was causal to fosfomycin resistance. Nonetheless, a few resistant *murA* alleles have previously been described. For example, purified MurA C115D or MurA C115E displayed resistance to fosfomycin in vitro[66]. It was suggested that, while these mutations to either Asp or Glu delete the target residue for covalent attachment of fosfomycin, they maintain the possibility to take over the role of Cys to act as the general acid in the enzyme-catalyzed reaction[66]. Indeed, although the activity of MurA was shown to be severely affected by the C115E mutation, MurA C115D retained high activity in vitro[66]. However, neither of these mutations were detected here. In fact, these mutations are completely absent from both replicates of the MurA saturation editing libraries, suggesting that they could not support viability in vivo. This finding highlights the potential differences between in vitro and in vivo behavior and stresses the need to investigate gene function as close to its natural context as possible, although it remains possible that both of these variants are missing from the library due to a combination of incomplete saturation levels and random chance. Additionally, two other *murA* mutations that provide resistance against fosfomycin were detected in clinical *E. coli* strains, D369N and L370I[67]. The MurA D369N mutation was initially present at very low levels in our library (Supplementary Data 1), but could no longer be detected when the library was grown for several additional generations (Supplementary Data 3) and was also not found when applying selective pressure by the adding fosfomycin. This could be due to the relatively high frequency of other spontaneous suppressor mutants that were preferentially isolated compared to MurA D369N and/or fitness defects associated with the D369N variant. The L370I mutation was not present in the library either because of insufficient saturation levels or because this mutant does not support viability in the *E. coli* lab strain used in this study. Taken together, we conclude that MurA is a very attractive target for new antibiotics because it cannot be easily mutated to overcome direct inhibition by antimicrobial compounds. Moreover, the MurA-fosfomycin combination is excellent in terms of resistance development through target modification. However, other mechanisms that provide resistance against fosfomycin exist and would have to be overcome to fully benefit from the powerful MurA-fosfomycin combination. Indeed, it is known that, in vitro, resistance to fosfomycin develops more easily through mutations that limit the import of this antibiotic into the cell than through mutations in the target MurA itself[57]. Thankfully, these import-limiting mutations are rare in vivo since they come with a considerable biological cost[68].

Likewise, we selected mutations in *lpxC* and *fabZ* that provide resistance against the LpxC-targeting compounds CHIR-090[58,59] and PF-04753299 (Pfizer). Surprisingly, all identified resistance-conferring mutations in *lpxC* are located in the N-terminal part of the protein. However, based on the experimentally determined structure of *Aquifex aeolicus* LpxC bound to CHIR-090[59], it seems highly unlikely that any of these mutated residues are directly involved in compound binding. Instead, since all the amino-acid substitutions we identify are located in the 5'-end of the gene, it is possible that they provide resistance by altering protein levels. All the identified mutations encode an amino-acid substitution and one or multiple synonymous PAM-site mutations. This change in codon usage at the start of the gene could alter expression levels by a variety of mechanisms, such as an altered speed of translation, changes in transcript stability or others[36]. Rather than influencing compound binding that occurs at an entirely different location, it therefore seems plausible that these 5'-end mutations would increase resistance levels by influencing protein production. Alternatively, the N-terminal domain could be involved in a previously undescribed regulatory mechanism that influences LpxC activity. It is perhaps unsurprising that no resistant *lpxC* mutations were found at the compound binding site. CHIR-090 is known to bind at the catalytic site[59] and we demonstrate that tolerance scores for residues found at or near this site are very low, meaning that not many mutations are tolerated and that therefore not many potentially resistance-conferring mutations are available at this location. To the best of our knowledge, no mutants with single-amino-acid substitutions in LpxC resistant to CHIR-090 or PF-04753299 have been reported previously.

Although CHIR-090 and PF-04753299 target LpxC, it is known that mutations in *fabZ* can provide resistance against anti-LpxC compounds[60–62]. Many such resistance mutations were detected throughout the entire FabZ protein. These include several residues that are also targeted in previously discovered mutants resistant to anti-LpxC compounds[60–62]. These *fabZ* mutations are believed to provide resistance against anti-LpxC drugs by lowering the activity of FabZ and thereby restoring the balance between phospholipid synthesis and LPS production[61]. It is therefore not surprising that so many different mutations in FabZ were isolated; any mutation that lowers FabZ activity appropriately is expected to provide resistance.

Apart from prioritizing potential antibiotic targets, we can also rank lead compounds based on the likeliness of resistance development. From our experiments using two anti-LpxC compounds, it became clear that PF-04753299 is superior to CHIR-090 from a resistance development point of view. Taken FabZ and LpxC together, there are less mutations – and importantly, less 1 SNP mutations – that confer resistance against PF-04753299 than CHIR-090. We therefore expect that, also in vivo, resistance is less likely to develop against PF-04753299, which is an important advantage for further drug development.

Taken together, we here present a deep mutational scanning approach that directly targets the *E. coli* genome and is able to interrogate the effect of selected mutations in vivo in its natural genomic context. We have used this approach to study the importance of individual amino acids in the function of three essential proteins involved in *E. coli* cell envelope synthesis. Additionally, we have exploited the CRISPR generated saturation editing libraries to formulate recommendations for antibiotic development based on predictions of the ease of resistance development. Our work may therefore contribute to future endeavors to select and validate targets for the development of new antibiotics.

## Methods

### Bacterial strains, compounds and growth conditions

Experiments were performed with *E. coli* SX43[69], a derivative of BW25993, except for CRISPR-FRT where *murA* mutations were transferred to *E. coli* BW25113 Δ*sfsB*[70]. Cultures were grown on/in SOB growth medium with/without 1.5% agar. They were incubated at 37 °C with continuous shaking at 200 rpm for liquid cultures, except for performing CRISPR-FRT which was done at 30 °C[71].

Compounds used include CHIR-090 (VWR International), PF-04753299 (Sigma-Aldrich) and fosfomycin (TCI Europe) at different concentrations, as indicated in the text. Additionally, carbenicillin (1000 μg/ml), chloramphenicol (68 μg/ml), gentamicin (25 μg/ml), kanamycin (40 μg/ml), spectinomycin (50 μg/ml) and anhydrotetracycline (100 ng/ml) were used where appropriate.

### Saturation editing library design and generation

Saturation editing libraries were constructed using high-throughput CRISPR-based editing provided by the Onyx® Digital Genome Engineering platform. Briefly, repair templates were designed using Inscripta's Designer software (development version) so that each amino acid would be replaced by every other amino acid and so that every codon would be replaced by a synonymous codon (if a synonymous codon exists). Besides the desired mutation, each oligo may also contain one or more synonymous edits that prevent re-cutting by eliminating the PAM site and/or introducing edits that interfere with cutting. For each mutation present in the repair template, the most frequently used available codon was chosen. 'Barcode Plasmids' containing the repair template, corresponding sgRNA and unique barcode, were cloned in bulk into a high-copy plasmid backbone. Three libraries were built each targeting a different gene (See Fig. 1).

### Cell library construction on Onyx®

Saturation editing libraries for *lpxC*, *fabZ* and *murA* were generated on the Onyx® Digital Genome Engineering Platform, a commercial benchtop instrument sold by Inscripta, Inc. Onyx® (Cat. #1001176) is a fully automated instrument that uses the MAD7 nuclease, a type V CRISPR nuclease from *Eubacterium rectale*, to generate multiplexed genome engineered libraries. All consumables, assays and software used in this study are available for purchase online at https://portal.inscriptacp.com/. Genome editing was performed using developmental reagents and protocols optimized for *E. coli* MG1655 (Onyx™ Engineering Handbook *E. coli* and *S. cerevisiae*. 2022. 1001178, https://inscripta.showpad.com/share/rWxQsFGmJznLKrdfWBHGH). A single *E. coli* SX43 colony was isolated from an LB agar plate and grown overnight in LB to OD600 2.5–4.1 mL of cell suspension was subsequently prepared using the Onyx® *E. coli* Edit Competency Kit (GEN-EC-1004). 1 mL of *E. coli* SX43 cells (approximately 6 × 10⁸ cells) prepared using the Edit Competency Kit were placed into the Onyx® instrument. The OnyxWare program K-strain v1.1 was selected and the Onyx® run was initiated. Briefly, the instrument transferred the cells to a cell growth cuvette (REF 1001155/Cat. #GEN-EC-1007) for growth to 0.5 OD, as measured on instrument. After an initial outgrowth, the instrument transferred cells to the microfluidic cell controller (REF 1001152/Cat. #GEN-EC-1007). There, cells were prepared for electroporation using media exchange. Once they were rendered competent, the instrument moved the cells to the microfluidic cell transformer (REF 1001152/Cat. #GEN-EC-1007), which controls introduction of the MAD7-containing engine plasmid as well as the gRNA/repair template/barcode-containing plasmid into cells by electroporation. Following electroporation, cells were placed by the instrument into a second cell growth cuvette (REF 1002161/Cat. #GEN-EC-1007) for recovery. Cells were then transferred to the digital engineering processor (REF 1001153/Cat. #GEN-EC-1007) for abundance normalization. The resulting normalized pool of cells was collected as multiple tubes from the instrument. 5 mL of cells per library were collected at an OD600 ranging from 5.4 to 7.6. Cells were immediately stored frozen at −80 °C in 15.5% glycerol. Depending on cell growth, total run time on instrument for *E. coli* SX43 lasted around 48 h.

For each experiment conducted with these libraries, 450 μL of thawed library material was used to begin an experiment. Following editing, the pooled libraries were grown off-instrument for approximately 8 h in LB supplemented with 1000 μg/ml carbenicillin and 68 μg/ml chloramphenicol. Both the gRNA-containing barcode plasmid and the MAD7-containing plasmid would be maintained with dual antibiotic selection. Library edit fractions were estimated using pooled whole-genome sequencing (pWGS) and ranged from 15–18% for these libraries[72]. Based on pWGS, we estimate that each experiment started with $2.9 \times 10^8$–$4.9 \times 10^8$ edited cells. Calculated as a per-edit coverage (based on libraries ranging in size from 2995 for *fabZ* to 8349 for *murA*, and assuming equal abundance in the pool), we estimate each surveyed edit would be covered by $3.5 \times 10^4$–$1.6 \times 10^5$ edited cells in the pool at the beginning of each experiment. A minimum transfer of $4.0 \times 10^7$ cells was maintained at all subsequent steps to avoid population bottlenecks.

Each Onyx® library was built as two biological replicates on independent *E. coli* SX43 isolates. Correlation between replicates was high according to the genomic amplicon assay described below; Spearman's ρ between 0.872 and 0.946 (Supplementary Fig. S1). Results presented here are from the first biological replicate.

### Illumina sequencing for detection of mutations in saturation editing libraries

Genomic DNA was isolated according to standard Inscripta protocols (Onyx™ Genotyping Handbook *E. coli* and *S. cerevisiae*. 2022. 1001182 RevB.), which uses the Wizard SV Genomic DNA Purification System (Promega, Cat. #A2365/A2360/A2361). PCR amplification of 2 kb genomic regions flanking each gene of interest (*fabZ*, *lpxC*, *murA*) was carried out in a reaction mixture containing 10 ng plasmid DNA, and 1 μl each of gene-specific forward and reverse primers at 20 μM (Supplementary Table S2) in a Q5 Hot-Start PCR Master Mix to a total volume of 50 μL. Cycling was carried out in BioRad T100 Thermal cycler instrument as follows: 98 °C for 2 min; 17 cycles of 98 °C for 10 s, 60 °C for 10 s, 72 °C for 1 min with a final extension at 72 °C for 5 min. PCR fragments were purified and prepared for sequencing according to the Inscripta Onyx™ Genotyping Handbook (Onyx™ Genotyping Handbook *E. coli* and *S. cerevisiae*. 2022. 1001182 RevB. https://inscripta.showpad.com/share/rWxQsFGmJznLKrdfWBHGH).

Amplified 2 kb genomic regions were sequenced as 150 bp paired end reads on an Illumina NextSeq. Sequencing data are made available at the Sequence Read Archive repository with BioProject accession number PRJNA887006. 54 to 66 million reads were collected per library. Expressed as per nucleotide coverage, coverages ranged from $8.14 \times 10^6$ to $9.97 \times 10^6$ per nucleotide were observed. Expressed as read coverage per design, coverages of $6.51 \times 10^3$ for *murA*, $10.96 \times 10^3$ for *lpxC* and $19.19 \times 10^3$ for *fabZ* were achieved. Overall, the number of 'Onyx edits' identified per design at this sequencing depth was 66.76 (±103.30). Expressed per library, the depths per design were: *murA*: 39.65 (±50.49), *lpxC*: 73.39 (±101.12), *fabZ*: 128.51 (±167.74).

Designs were quantified using Inscripta's proprietary genomic amplicon edit detection pipeline. Briefly, this approach performed competitive alignment to determine the origin of each read. Each read was aligned to an augmented reference, which contained the original unedited reference genome, supplemented by a set of alternative contigs. The alternative contigs consisted of, for each design, the repair template (corresponding to the Onyx® edit) in its genomic context and in the cassette backbone context. Alignment was performed using BWA-MEM (version 0.7.17)[73]. Ambiguously mapped reads were remapped to the unedited reference. Each read was categorized as either providing evidence for the complete intended edit (Onyx® edit reads), no edit ('reference' reads), or insufficient evidence to make a call. Importantly, an 'Onyx® edit' consists of both the intended non-synonymous variant as well as any required additional 'immunizing edits' that prevent MAD7 recutting by modifying the PAM sequence with synonymous edits (if the PAM occurs in a gene CDS). Thus, false positive 'Onyx® edit' calls are unlikely to result from rare Illumina sequencing errors, as the error would have to occur in multiple, non-contiguous nucleotides.

### Library selection experiment and analysis

Each gene library was grown in triplicate over three growth cycles in SOB supplemented with 1000 μg/ml carbenicillin at 37 °C for a total of 15 generations. Under these selection conditions, only the gRNA-containing barcode plasmid would be maintained. Cultures were continuously kept at OD ≤ 0.5 by appropriately timed dilutions. After 0, 5 and 15 generations of growth, 2 kb genomic amplicons from the cell population were sequenced to quantify changes in the abundance of the edits over time. A weighted least squares log-linear model was fit to model the change in abundance of each mutation using Enrich2 (version 1.3.1)[42]. Regression weights were calculated by Enrich2 according to the Poisson variance of each mutation's read counts across replicates, accounting for variable measurement error rates that depend on sequencing depth. The slope of the linear fit was taken to be a mutation's competition coefficient. Competition coefficients were normalized to the total read counts in the sample and adjusted such that the most common competition coefficient (mode) was set to 0. Competition coefficient values greater than 0 indicate improved growth, whereas coefficients less than 0 indicate impaired growth. Mutations with fewer than 5 reads at the beginning of the experiment were excluded from the analysis.

### Calculating tolerance scores

The tolerance scores were calculated by using a modified version of the Zvelebil similarity score[43], which is based on counting key differences between amino acids. For each sequence position, the score was calculated for each tolerant non-synonymous mutation using the characteristics 'small', 'aliphatic', 'proline', 'negative', 'positive', 'polar', 'hydrophobic' and 'aromatic', whereby for each difference in each of these characteristics between the original and mutated amino acid, a score 0.1 was added to a starting score of 0.1 (effectively a reversal of the original Zvelebil score). Per sequence position, all scores for tolerant mutations were summed, then normalized by dividing by the maximum difference score possible for the original amino acid (the score obtained when mutated to all other possible amino acids). A score of 1.0 therefore indicates full tolerance in that position, a score of 0.0 no tolerance, and higher in-between scores increasing levels of tolerance for that amino acid type, with mutations to dissimilar amino acids contributing more.

### Protein structures & RSA values

Protein structures shown and used in this manuscript were obtained from the PDB, with the codes and corresponding references for the used protein structures: FabZ PDB 6n3p, chain A[55], LpxC PDB 4mqy, chain A[44], MurA PDB 1uae, chain A[38]. RSA (Relative Solvent Accessibility) values were extracted from these single chain protein structures using the PoPMuSiC software[74]. PDB structures colored by tolerance score (Fig. 6 and Supplementary Data 7) were obtained using PyMOL Molecular Graphics System, Version 2.5.0.

### MIC tests

MIC tests were performed according the broth dilution method in SOB medium. Briefly, OD 625 nm of overnight cultures was adjusted to 0.1. Cells were then diluted 200 times and 2-fold dilution series of the tested compounds were added. Cells were incubated for 24 h, after which OD 595 nm was measured. MIC values were chosen as the lowest concentration of added compound that led to OD 595 nm values <10% of OD 595 nm values of the untreated control. For all MIC tests, 3 biological repeats were performed, each consisting of 2 technical replicates per condition. The MIC value most frequently encountered and/or centered in between all detected values was chosen as the final MIC. MIC values of *E. coli* SX43 for CHIR-090, PF-04753299 and fosfomycin were determined to be 0.032 μg/ml, 0.5 μg/ml and 8 μg/ml, respectively.

### Selection and identification of resistant variants

To select variants from the FabZ, LpxC or MurA libraries resistant to CHIR-090, PF-04753299 or fosfomycin, various amounts of frozen library stocks were plated onto SOB agar plates containing one of these compounds at a concentration of 4x, 8x or 32x MIC. After overnight incubation, colonies that were able to form under these conditions were counted to determine the resistant CFUs/ml to each different concentration. Simultaneously, library stocks were also plated on non-selective SOB agar plates to determine total cell concentrations in each of the stocks. The number of resistant CFUs/ml was then normalized to the total cell concentration of the library to determine the frequency of resistance. Additionally, to compare the occurrence of resistance between the saturation editing libraries and a wild-type strain, various amounts of a wild-type SX43 culture were plated onto non-selective SOB plates and plates containing 4x, 8x or 32x MIC of the various compounds. After overnight incubation, resistant CFUs/ml to each different concentration were determined by counting colonies on selective plates. These numbers were normalized to the total cell concentration in the overnight culture.

To confirm that colonies able to form on selective plates after overnight incubation are indeed resistant, they were transferred to the wells of a microtiter plate containing 200 μl SOB medium supplemented with the compound at the concentration used for initial selection of the specific clones. If clones were able to grow overnight, they were deemed resistant with an MIC value higher or equal to the concentration of compound used for selection of this strain.

To identify mutations in these resistant strains, their *lpxC*, *murA* or *fabZ* gene (dependent on the library the clones were isolated from) was amplified by PCR using primer pairs SPI12880 & SPI12881, SPI12882 & SPI12883 or SPI12884 & SPI12885, respectively (Supplementary Table S2). PCR products were sent for Sanger sequencing using the same primers that were used for amplification.

Due to the absence of mutations in *murA* for the MurA library selected on fosfomycin, resistance of all selected clones was again confirmed with a MIC test (only 1 biological repeat performed). All isolated clones displayed an increase in MIC that was equal to or greater than the concentration of fosfomycin on which they were selected.

### CRISPR-FRT to transfer mutations to new genetic background

Selected *murA* mutations were transferred to a clean genetic background using CRISPR-FRT[71], a modified CRISPR-Cas protocol that targets FRT sites present in the *E. coli* Keio library. The Keio library is a collection of around 4000 mutants that all contain a different single-gene deletion where the gene in question is replaced by

an FRT-KmR-FRT cassette[70]. By targeting FRT sites using CRISPR-Cas and providing a rescue oligo that contains flanking homologous region, the deleted gene can be replaced with any desired sequence present on the rescue oligo in between the homologous regions. Selected *murA* mutant alleles, together with extended up- and down-stream regions, were amplified from selected resistant variants using primers SPI14680 & SPI14681. These PCR products were used as rescue oligos. They were transformed to an *E. coli* BW25113 Δ*sfsB* mutant that contains the CRISPR-Cas editing plasmids pCas9CR4-Gm and pKDsgRNA-FRT[71]. *sfsB* is a non-essential gene just downstream of the essential *murA* for which no deletion mutant is present in the Keio library. CRISPR-Cas editing was performed as described before[71] and colonies were selected for their loss of kanamycin resistance. Colony PCR on the *murA* gene of selected clones was performed using primers SPI12882 & SPI12883 and PCR products were sequenced with the same primers to confirm the presence of the transferred *murA* mutations.

### Reporting summary

Further information on research design is available in the Nature Portfolio Reporting Summary linked to this article.

### Data availability

Sequencing datasets generated during the current study have been deposited in the Sequence Read Archive repository with BioProject accession number PRJNA887006. PDB files used in this study are FabZ PDB 6n3p, LpxC PDB 4mqy, and MurA PDB 1uae. Read counts, other processed data resulting from our sequencing datasets, and PDB files (not generated during this study) displaying tolerance scores (calculated in this study) and corresponding PyMol scenes are provided as Supplementary Data. The MIC and resistance data generated in this study are provided in Supplementary Data and as Source Data file. Source data are provided with this paper.

### Code availability

Though the code for the genomic amplicon pipeline is part of Inscripta's proprietary software, the details required for reimplementation using public software have been fully described in the methods. Custom Python (version 3.9 & 3.10.8) and R (RStudio version 1.4.1106) analysis scripts for data analysis will be made available upon request.

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

## Acknowledgements

The work was supported by grants from the Research Foundation Flanders (FWO) (G0B0420N, G055517N, G0C4322N, G0I1522N), KU Leuven (C16/17/006), Vrije Universiteit Brussel (SRP50), Francqui Research Foundation, VIB Technology Watch and VIB. L.D. received an FWO postdoctoral fellowship.

## Author contributions

Conceptualization: L.D., A.N.B., K.N., N.K., W.Versées, W.Vranken, J.M.; Methodology: L.D., A.N.B., K.N., N.K., W.Versées, W.Vranken; Formal analysis: L.D., A.N.B., W.Versées, W.Vranken; Investigation: L.D., A.N.B., K.N., C.C., A.C.E., T.S.; Writing – Original Draft: L.D., W.Versées, W.Vranken; Writing – Review & Editing: L.D., A.N.B., K.N., N.K., W.Versées, W.Vranken, J.M.; Visualization: L.D., A.N.B., W.Versées, W.Vranken.

## Competing interests

A.N.B., K.N., C.C., A.C.E. and N.K. are affiliated with Inscripta, Inc. The remaining authors declare no competing interests.
