## [Peer Review File · Nature Communications]

Deep mutational scanning of essential bacterial proteins can guide antibiotic developmentREVIEWER COMMENTS

Reviewer #1 (Remarks to the Author):

This is an interesting paper that describes the use of deep mutational scanning of three essential *E. coli* enzymes (FabZ, LpxC, and MurA) involved in cell wall biosynthesis. There have been many deep mutational scanning projects published on various proteins. However, the interesting aspect of this paper is that the codon randomization libraries were constructed using CRISPR in the gene that resides in its natural position in the chromosome rather than a cloned version on a plasmid, as is commonly done. Therefore, the gene is under its normal regulation and copy number. Based on the mutagenesis results, the authors conclude that MurA is a superior antibiotic target because it is less tolerant to mutations than the other enzymes and thus has more constraints on the evolution of drug resistance. The work is innovative in the sense that the mutations are introduced into the chromosome and the use of synonymous codons allows an estimation of the level of mutagenesis achieved in the libraries.

The paper is well-written and provides new insights into the *in vivo* sequence requirements for these enzymes. In addition, the use of the saturation libraries to examine resistance mutations is a valuable assay for resistance mutations and revealed hotspots for resistance.

A surprising finding is the number of mutations that are tolerated and still pass the selection, i.e., they are viable. This is particularly true for FabZ, which has very relaxed sequence requirements, even for residues presumed to be catalytically important. This raises the question of what level of enzyme activity is required for viability. For example, if only 1% of FabZ activity is required for viability while 50% of enzyme activity is required for LpxC then the interpretation of the results would be different for each. It would be useful to purify a few mutants that are selected as viable or at least a representative one from each enzyme class and measure its catalytic activity to get an idea of how much activity is associated with the selected viable mutants. In addition, it would be of interest to passage the libraries of mutants for multiple generations by passaging the cultures through multiple overnight growths to see if this distinguishes mutants that have small differences in fitness.

Reviewer #2 (Remarks to the Author):

This paper demonstrates a great use of a powerful technology, high-throughput CRISPR genome editing for deep mutational scanning of essential genes in *E. coli* genome. As the author argued, to do most DMS targeted genes located in plasmids and needs to have some technical struggle to interpret data because of different copy numbers etc. The technology that the authors presented is elegant and powerful to study mutational effects directly on genome, but generated targeted library in one gene.

The demonstration of the technology is very exciting, however, I have to say that the execution, data analysis and presentation of DMS are extremely poor and they are far from acceptable from my point of view. As DMS is very noisy experiment, the authors have to some control experiments, proper data analysis and present all data in the manuscript. Currently, there is essentially none, there is no way that other researchers can replicate their work at all. Therefore, I believe that the paper does not fulfil a minimum standard to publish and I need to suggest to reject the manuscript with the current format. I have listed points that the authors can address to provide quality of data and presentation for DMS.

1. There is no numbers of actual library size for each protein. How many *e. coli* colonies are handled after library construction and culture? Which should tell what would a coverage of variants (theoretical variants/actual cell numbers).

2. What is the coverage of Illumina sequencing for each library? How many time each position has been read? Those numbers are important to ensure non-observed variants can be considered as truly deleterious mutations or just lack of the good coverage of the library.

3. Have the authors filtered and quality checked potential mutations during Illumina sequencing?

Some mutations (e.g., 0.1%) are expected from sample preparation and Illumina sequencing as the authors directly sequencing targeted genes. It sounds like the authors consider all observed mutations as survival mutations (I cannot really judge as there is no description of quality control and data analysis etc.). I suspect that some variants that the authors consider tolerated can be from Illumina sequencing. There is many ways to quality control those errors, Illumina sequence WT gene and estimate mutation rates from sample preparation and sequencing, and/or simply remove low counts signals. The authors should share all data including Illumina raw counts and processed data in supplementary file too (there is none at the moment). Without proper quality controls and descriptions, I cannot simply just what they are claiming is true or not.

4. Have the authors performed independent DMS experiments at least twice (culture, sample prep and sequencing)? I would like to see everyone does it to ensure the experiment is running fine and present correlation plots between biological replicates. It is extremely important as DMS generates lots of data and the authors must ensure the quality is good and present those quality test properly.

5. There is no follow up experiment that DMS is correct in this paper. The authors should pick a handful of variants and make sure that each individual strain can grow as the WT strain. The argument to compare previously published mutations are good, but it is important to perform validation experiments as DMS is yet noisy experiments.

6. Why the author did not do Illumina sequencing for Fosfomycin resistance variants? It really does not make sense as the authors possess a technique to do. They need to sequence variants before and after the selection. Many variants were identified only once by Sanger sequencing which does not give much confidence. I feel that the work lost the impact they authors did not use their maximum ability to analyze the library they constructed.

Reviewer #3 (Remarks to the Author):

This very interesting manuscript uses deep mutational scanning studies to determine the tolerance to mutation of three essential E. coli genes in their native loci. One of the more interesting findings was the positive correlation between mutational tolerance and frequency of antibiotic resistance mutations in these three genes. I have several comments that I hope the authors will find useful in improving the manuscript. These mostly concern how the data is analyzed and presented and possible alternative explanations of the high mutational tolerance the authors observe as well as contradictions the authors found with previous studies.

Major comments

1. Line 110 "The Onyx technology optimizes for maximum representation of variants during library construction to prevent skews in the population due growth competition. We also restricted growth to what was needed for library construction (see Materials & Methods) to limit competition between constructed variants and retain all variants that support viability." I found it difficult to evaluate the first sentence because the Materials and Methods mostly just refers to a series of handbooks on the Onyx system. I think the details are important here and all the steps from transformation until isolating the plasmid should be explicitly described in the methods rather than referring to online manuals that a) have a number of different alternatives described (i.e. what precisely did the authors do? Were any plasmids cured? etc.) and b) might not be available online in the future.

2. Figure 2A-C. I highly recommend arranging the amino acids in a more logical order instead of alphabetical. By grouping similar amino acids together, patterns are more easily observed. There is no single best way to do this grouping. I suggest using whatever grouping the authors prefer after looking at what other researchers have done when reporting heat maps of deep mutational scanning data.

3. Figure 2AC color scheme is highly biased toward showing any non-lethal mutation as a hue of blue,

whereas mutations with just a 50% reduction in fitness are highly deleterious from an evolutionary point of view. This coloring scheme serves the authors' conclusion of showing the proteins are highly tolerant to mutation. However, a mutation that is highly depleted because of significant fitness costs is not a tolerated mutation even if the authors manage to observe it. This difference in perspective should be made clear by the authors. I believe the difference stems from the authors' focus on antibiotic resistance, where antibiotics need to completely kill the bacteria...thus the focus on mutations that are completely deleterious (selection coef ≤ -1).

4. Figure 2 legend "The frequency of occurrence was normalized to the sum of the frequency of occurrence of all mutations at the same position and is indicated by a blue color scale." What is the rationale for the reference being the sum of the frequency of occurrence at that position? A mutation that is perfectly tolerated might get normalized to 50% if it and the wildtype are the only tolerated amino acid, or it might get normalized to 5% if all mutations are completely tolerated. Thus, there is the possibility of a 10-fold difference in the measure of tolerance for two mutations that are both completely neutral. Wouldn't it be better to normalize to the frequency of the synonymous mutation at that position, which serves as the control for neutral mutation creation efficiency?

5. Figure 2D-F. Whether a mutation is going to count as "successful" or not (as the authors are defining it) will depend on three things 1) the frequency at which that mutation was introduced into the genome, 2) the effects of that mutation on cell fitness, and 3) the number of generations of growth of the library after the mutation is introduced. The authors state they have tried to minimize #3, presumably so that they could observe any mutation that has a selection coefficient > -1 , even if the fitness cost was substantial. However, some growth is presumably necessary to enrich for tolerated mutations so that the mutation is not observed for cells that do not grow because the mutation is lethal. If this is the authors' intention, how do they know they have the right balance in the number of generations of growth? Also, the authors' approach does not seem to account for #1. For example, consider position A where mutations are made with great frequency and position B where mutations are made at low frequency. If the two positions were equally tolerant to mutation (but some mutations have deleterious fitness effects), using the authors' approach, position B would appear less tolerant. I would suggest that the authors are ignoring useful information in plotting Fig 2D-E: the frequency at which the mutations are observed relative to the synonymous mutation. The authors view it binary (did the mutation appear or not). Perhaps instead (or in addition) the authors could sum up the frequencies relative to the synonymous mutation at that position (see my comment #4 above). That way if all mutations at a position reduced fitness by 90%, you would get a low value (instead of a 19 the way the authors are doing it).

6. The authors note surprising observations of mutations that are tolerated that seem to contradict previous studies on these proteins. In addition, the authors found FabZ to be remarkably tolerant to mutation given its essential nature. One possibility the authors do not consider is the frequency of mistranslation. Codons in an mRNA message are prone to mistranslation errors in ways that depend on the mutant codon, the neighboring codons, and the bacterial strain being used. Perhaps the level of active protein necessary for the authors to observe a mutation and mark it as successful is very low (say 1% or 0.1% of the wildtype level). This can happen if fitness is greatly buffered to changes in total cellular protein activity, which has been observed for some genes. Thus, mistranslation to incorporate the wildtype (or another tolerated) amino acid at a mutated position would not need to be very efficient for that mutation to appear tolerated even though a protein with that mutation is inactive, for example. I'm not suggesting this possibility seriously impacts the authors' findings, after all the mutation is still tolerated regardless of the mechanism. However, I think the authors should discuss whether mistranslation might impact or explain some of their results, especially the apparent contradictions with previous studies (e.g. in vivo vs. in vitro differences).

Minor

1. Line 130. Extra period

Response to reviewer comments

Reviewer #1 (Remarks to the Author):

This is an interesting paper that describes the use of deep mutational scanning of three essential *E. coli* enzymes (FabZ, LpxC, and MurA) involved in cell wall biosynthesis. There have been many deep mutational scanning projects published on various proteins. However, the interesting aspect of this paper is that the codon randomization libraries were constructed using CRISPR in the gene that resides in its natural position in the chromosome rather than a cloned version on a plasmid, as is commonly done. Therefore, the gene is under its normal regulation and copy number. Based on the mutagenesis results, the authors conclude that MurA is a superior antibiotic target because it is less tolerant to mutations than the other enzymes and thus has more constraints on the evolution of drug resistance. The work is innovative in the sense that the mutations are introduced into the chromosome and the use of synonymous codons allows an estimation of the level of mutagenesis achieved in the libraries.

The paper is well-written and provides new insights into the *in vivo* sequence requirements for these enzymes. In addition, the use of the saturation libraries to examine resistance mutations is a valuable assay for resistance mutations and revealed hotspots for resistance.

A surprising finding is the number of mutations that are tolerated and still pass the selection, i.e., they are viable. This is particularly true for FabZ, which has very relaxed sequence requirements, even for residues presumed to be catalytically important. This raises the question of what level of enzyme activity is required for viability. For example, if only 1% of FabZ activity is required for viability while 50% of enzyme activity is required for LpxC then the interpretation of the results would be different for each. It would be useful to purify a few mutants that are selected as viable or at least a representative one from each enzyme class and measure its catalytic activity to get an idea of how much activity is associated with the selected viable mutants. In addition, it would be of interest to passage the libraries of mutants for multiple generations by passaging the cultures through multiple overnight growths to see if this distinguishes mutants that have small differences in fitness.

We thank the reviewer for this comment. Purifying mutated enzymes and determining their catalytic activity *in vitro* is however a very elaborate and time-consuming experiment, especially considering that many of the substrates needed are not commercially available and would therefore have to be synthesized by us for *in vitro* testing. We currently do not feel able to take on such an intense and elaborate experiment. However, as was suggested by the reviewer, previously published results do indicate that the level of FabZ enzyme activity required for viability is somewhat lower. CRISPRi experiments have shown that growth is more robust against depletion of FabZ than LpxC and MurA (Donati *et al.* 2021 Cell Systems), meaning that FabZ's catalytic activity could be present in excess in the cell in comparison to LpxC or MurA. These findings were added to our manuscript at line 163.

Additionally, we agree with the reviewer that ascertaining fitness consequences of the mutations would strengthen our analysis and help identify mutations that are tolerated but have lower fitness as a result of their impact on protein function. As suggested by the reviewer, gene libraries were reconstructed and grown, in triplicate, for 15 generations in total. Cultures were continuously kept at $OD \leq 0.5$ by appropriately timed dilutions. After 0, 5 and 15 generations of growth, we sequenced the cell population to quantify changes in the abundance of the edits over time. By fitting a log-linear model (using a software tool called Enrich2; Rubin *et al.* 2017 Genome Biology) we were able to estimate competition coefficients for each edit (slope of the log-linear model). Using this approach, we were able to characterize fitness effects associated with detected edits. For example, we now estimate that hundreds

of mutations confer a “significant fitness burden” (*fabZ*: 107 [4.7%]; *lpxC* 161 [3.29%]; *murA* 164 [2.79%]; gene #mutations [%total mutations]), where significant fitness burden is defined as mutations with a competition coefficient \leq two standard deviations from the most common coefficient, 0 (Figure S4A). These results are a valuable addition to our manuscript and are presented in Results paragraph “Competition within saturation editing libraries reveals the fitness effect of each edit” (starting at line 148), Tables S3 and S4 and Figures 2, 4, S2, S3 and S4.

Importantly, however, these experiments showed that while our original optimized approach could indeed reliably identify the vast majority of important residues in LpxC and MurA, this was not the case for FabZ. The data from the selection experiment demonstrated that FabZ variants require additional generations of growth for depletion of harmful edits to occur (Figure 2). Because of these findings, and to remain consistent throughout our manuscript, we have now performed our analyses on FabZ, LpxC and MurA libraries that were grown for an additional 15 generations. Whereas the identified important residues for LpxC and MurA remain the same, our approach is now able to discover more FabZ residues that are important for protein function. We want to thank the reviewers for their comments that have prompted us to look into the FabZ library in more detail, because we strongly believe that the new analyses presented in our revised manuscript represent a significant improvement to our work.

Reviewer #2 (Remarks to the Author):

This paper demonstrates a great use of a powerful technology, high-throughput CRISPR genome editing for deep mutational scanning of essential genes in *E. coli* genome. As the author argued, to day most DMS targeted genes located in plasmids and needs to have some technical struggle to interpretate data because of different copy numbers etc. The technology that the authors presented is elegant and powerful to study mutational effects directly on genome, but generated targeted library in one gene.

The demonstration of the technology is very exciting, however, I have to say that the execution, data analysis and presentation of DMS are extremely poor and they are far from acceptable from my point of view. As DMS is very noisy experiment, the authors have to some control experiments, proper data analysis and present all data in the manuscript. Currently, there is essentially none, there is no way that other researchers can replicate their work at all. Therefore, I believe that the paper does not fulfil a minimum standard to publish and I need to suggest to reject the manuscript with the current format. I have listed points that the authors can address to provide quality of data and presentation for DMS.

We appreciate the reviewer’s candid feedback and acknowledge that, in its original form, our manuscript did not contain enough experimental details to be able to judge the quality of the data collection or analysis. However, we want to stress that many of the issues raised by the reviewer were already taken into account in our original manuscript, but admittedly not explained sufficiently. In the revised version of our manuscript, we now incorporate detailed information that will allow the reader to more accurately judge the quality of our work. Below, we give more details on the changes made addressing each of the reviewer’s comments.

1. There is no numbers of actual library size for each protein. How many *e. coli* colonies are handled after library construction and culture? Which should tell what would a coverage of variants (theoretical variants/actual cell numbers).

With Inscripta's Onyx® platform, libraries are collected and handled as pools, rather than individual colonies. The point raised by the reviewer regarding library size is nonetheless critical for interpretation and we thank the reviewer for raising it. Based on OD600 measurements and edit fraction estimates from pooled whole genome sequencing (pWGS; described in detail in the expanded Materials & Methods section) we estimate that each experiment started with $2.9 \times 10^8 - 4.9 \times 10^8$ edited cells (total cells = $1.9 \times 10^9 - 2.7 \times 10^9$). Calculated as a per-design coverage (based on libraries ranging in size from 2995 for *fabZ* to 8349 designs for *murA*), on average we estimate a surveyed edit would be represented by $3.5 \times 10^4 - 1.6 \times 10^5$ edited cells in the pool at the beginning of each experiment. A minimum transfer of 4.0×10^7 cells was maintained at all subsequent steps to avoid population bottlenecks. These details are now disclosed in the methods section beginning at line 619.

2. What is the coverage of Illumina sequencing for each library? How many times each position has been read? Those numbers are important to ensure non-observed variants can be considered as truly deleterious mutations or just lack of the good coverage of the library.

We thank the reviewer for identifying these important details, which we had neglected to report. The number of reads mapped to the 2kb genomic amplicon ranged from 54 to 66 million per library. Coverage estimates can be made assuming even coverage across nucleotides and designs. Importantly, these estimates include both the edited and un-edited cell populations that comprise Onyx® cell libraries.

Expressed as per nucleotide coverage (given that these were 150 nt paired end reads), coverages ranged from 8.14×10^6 to 9.97×10^6 per nucleotide. Expressed as 'read coverage per design', coverages of 6.51×10^3 for *murA*, 10.96×10^3 for *lpxC* and 19.19×10^3 for *fabZ* were achieved. Overall, the number of 'Onyx edits' identified per design at this sequencing depth was 66.76 (± 103.30). Expressed per library, the depths per design were: *murA*: 39.65 (± 50.49), *lpxC*: 73.39 (± 101.12), *fabZ*: 128.51 (± 167.74). As described below, 'Onyx edits' are reads that include both the intended non-synonymous edit and any "immunizing edits" required to prevent MAD7 recleavage. These details are now disclosed in the methods section beginning at line 677.

3. Have the authors filtered and quality checked potential mutations during Illumina sequencing? Some mutations (e.g., 0.1%) are expected from sample preparation and Illumina sequencing as the authors directly sequencing targeted genes. It sounds like the authors consider all observed mutations as survival mutations (I cannot really judge as there is no description of quality control and data analysis etc.). I suspect that some variants that the authors consider tolerated can be from Illumina sequencing. There are many ways to quality control those errors, Illumina sequence WT gene and estimate mutation rates from sample preparation and sequencing, and/or simply remove low counts signals. The authors should share all data including Illumina raw counts and processed data in supplementary file too (there is none at the moment). Without proper quality controls and descriptions, I cannot simply just what they are claiming is true or not.

Thanks to the way our genomic edits are designed and the analysis pipeline used, we can be sure that the variants identified by Illumina sequencing are those that we have introduced by CRISPR genome editing and are not due to sequencing errors. This is because, apart from the desired edit, we always also introduce at least one additional synonymous mutation that is aimed at eliminating the PAM-site and thereby prevents re-cutting of the edited genomic site ("immunizing edit"). We only consider a desired edit present if we can detect all expected mutations (the desired edit and all associated immunizing edits). Sequencing reads that only partially match with the expected pattern are disregarded and removed from the analysis. Any sequencing errors would therefore be filtered out using this approach.

We realize that this has not been made sufficiently clear in the original version of this manuscript. We have therefore added a more extensive explanation in our revised version. This explanation can be found starting at line 696.

Additionally, all raw and processed sequencing data have been made available either as supplementary material for this manuscript or on the Sequence Read Archive (SRA) repository with BioProject accession number PRJNA887006 (reviewer link: <https://dataview.ncbi.nlm.nih.gov/object/PRJNA887006?reviewer=saa50dovc4ind37t37hlm5oect>).

4. Have the authors performed independent DMS experiments at least twice (culture, sample prep and sequencing)? I would like to see everyone does it to ensure the experiment is running fine and present correlation plots between biological replicates. It is extremely important as DMS generates lots of data and the authors must ensure the quality is good and present those quality test properly.

Replication is a critical aspect of any high-throughput experiment, including the one enabled by Onyx[®] presented here. We thank the reviewer for raising this important point. While experiments establishing biological replicability have been performed for the Onyx[®] technology, they have not been performed using direct edit detection with a genomic amplicon as described here. We therefore now built and sequenced independent Onyx[®] cell libraries for each gene. Molecular reagents (synthesized oligos) for the libraries were re-amplified and built on an independent *E. coli* SX43 colony using the described Onyx[®] on-instrument protocol. 2kb genomic amplicons from the resulting cell libraries were subsequently amplified and sequenced as described. We observed high-correlation between replicates (Spearman's ρ for FabZ 0.946, LpxC 0.900 and MurA 0.872). Replication of each gene library is presented in Figure S1 and the results are described in the main text (line 114) and methods section starting at line 658.

5. There is no follow up experiment that DMS is correct in this paper. The authors should pick a handful of variants and make sure that each individual strain can grow as the WT strain. The argument to compare previously published mutations are good, but it is important to perform validation experiments as DMS is yet noisy experiments.

To address the reviewer's concern, we have now reconstructed the libraries and performed a competition experiment where each library was grown for 15 generations. The composition of the library was followed by Illumina sequencing and competition coefficients were assigned to each edit to capture their potential fitness benefits (competition coefficient > 0) or defects (competition coefficient < 0).

As expected, many variants decreased in abundance, meaning that edits in these variants are associated with fitness defects. However, as is clear from Figure S4A (also shown below), the majority of variants display neutral competition coefficients and are therefore expected to grow as well as the wildtype.

The results obtained from this competition experiment are a valuable addition to our manuscript and are presented in Results paragraph "Competition within saturation editing libraries reveals the fitness effect of each edit" (starting at line 148), Tables S3 and S4 and Figures 2, 4, S2, S3 and S4.

Figure S4A: The distribution of competition coefficients for each library shows that the majority of edits have neutral competition coefficients and are therefore not associated with fitness defects.

6. Why the author did not do Illumina sequencing for Fosfomycin resistance variants? It really does not make sense as the authors possess a technique to do. They need to sequence variants before and after the selection. Many variants were identified only once by Sanger sequencing which does not give much confidence. I feel that the work lost the impact they authors did not use their maximum ability to analyze the library they constructed.

To be fully sure that selected mutants are resistant to compounds tested, we wanted to check and confirm each of them individually. Since we were then working with clonal populations, we figured it was easier to sequence them by Sanger than pool them again for Illumina sequencing. Moreover, this gave us the additional advantage of knowing exactly which *fabZ/lpxC/murA* mutation was present in each strain, which will prove useful for further follow-up studies.

Sanger sequencing of our obtained clonal populations may not be the most powerful approach, but we are convinced that it was capable of extracting the information that was needed to reach the conclusions presented in our manuscript.

Reviewer #3 (Remarks to the Author):

This very interesting manuscript uses deep mutational scanning studies to determine the tolerance to mutation of three essential *E. coli* genes in their native loci. One of the more interesting findings was the positive correlation between mutational tolerance and frequency of antibiotic resistance mutations in these three genes. I have several comments that I hope the authors will find useful in improving the manuscript. These mostly concern how the data is analyzed and presented and possible alternative explanations of the high mutational tolerance the authors observe as well as contradictions the authors found with previous studies.

Major comments

1. Line 110 “The Onyx technology optimizes for maximum representation of variants during library construction to prevent skews in the population due growth competition. We also restricted growth to what was needed for library construction (see Materials & Methods) to limit competition between constructed variants and retain all variants that support viability.” I found it difficult to evaluate the first sentence because the Materials and Methods mostly just refers to a series of handbooks on the Onyx system. I think the details are important here and all the steps from transformation until isolating the plasmid should be explicitly described in the methods rather than referring to online manuals that a) have a number of different alternatives described (i.e. what precisely did the authors do? Were any plasmids cured? etc.) and b) might not be available online in the future.

We agree with the reviewer that we should add these experimental details in the manuscript itself, which is why we have significantly extended the materials & methods section to include all needed information to appreciate the workflow of library generation. This explanation can be found starting at line 619.

2. Figure 2A-C. I highly recommend arranging the amino acids in a more logical order instead of alphabetical. By grouping similar amino acids together, patterns are more easily observed. There is no single best way to do this grouping. I suggest using whatever grouping the authors prefer after looking at what other researchers have done when reporting heat maps of deep mutational scanning data.

We have changed the grouping of amino acids of this figure into the following order: Polar acidic (D, E), Polar basic (K, R, H), Polar neutral (S, T, C, Q, N), Aromatic (F, Y, W) and Hydrophobic (G, A, V, L, I, M, P). We fully agree that this ordering makes more sense and aids in data interpretation.

3. Figure 2AC color scheme is highly biased toward showing any non-lethal mutation as a hue of blue, whereas mutations with just a 50% reduction in fitness are highly deleterious from an evolutionary point of view. This coloring scheme serves the authors' conclusion of showing the proteins are highly tolerant to mutation. However, a mutation that is highly depleted because of significant fitness costs is not a tolerated mutation even if the authors manage to observe it. This difference in perspective should be made clear by the authors. I believe the difference stems from the authors' focus on antibiotic resistance, where antibiotics need to completely kill the bacteria...thus the focus on mutations that are completely deleterious (selection coef ≤ -1).

It is true that our coloring scheme was biased towards highlighting non-lethal mutations. This was a conscious choice from our part because the frequency with which we detect genomic variants does not necessarily reflect their fitness. This frequency will depend on a variety of factors including their fitness, but also the frequency of each design cassette in the input pool for transformation, the efficiency of the targeted PAM site, and perhaps still other unknown factors. We therefore do not feel confident directly tying variant read counts to fitness.

However, we agree with the reviewer that ascertaining fitness consequences of the mutations would strengthen our analysis and help identify mutations that are tolerated but have lower fitness as a result of their impact on protein function. We therefore grew each gene library, in triplicate, for 15 generations in total. Cultures were continuously kept at $OD \leq 0.5$ by appropriately timed dilutions. After 0, 5 and 15 generations of growth, we sequenced the cell population to quantify changes in the abundance of the edits over time. By fitting a log-linear model (using a software tool called Enrich2; Rubin et al. 2017 Genome Biology) we were able to estimate competition coefficients for each edit (slope of the log-linear model). Using this approach, we were able to characterize fitness effects associated with detected edits. For example, we now estimate that hundreds of mutations confer a "significant fitness burden" (*fabZ*: 107 [4.7%]; *lpxC* 161 [3.29%]; *murA* 164 [2.79%]; gene #mutations [%total mutations]), where significant fitness burden is defined as mutations with a competition coefficient \leq two standard deviations from the most common coefficient, 0 (Figure S4A). These results are a valuable addition to our manuscript and are presented in Results paragraph "Competition within saturation editing libraries reveals the fitness effect of each edit" (starting at line 148), Tables S3 and S4 and Figures 2, 4, S2, S3 and S4.

With these extra data at hand, we have decided to change what was previously Figure 2A-C and rework them into Figures 4A-C that address the reviewer's comment. The color gradient in this figure now shows the read counts associated with each edit at the end of the selection experiment, i.e. after 15 generations of growth, normalized to the read counts at the start of

the experiment (0 generations). Values close to 0 indicate edits that confer significant fitness defects and can be seen as light shades of blue on the figure, while values of 1 and higher are displayed in dark blue.

4. Figure 2 legend “The frequency of occurrence was normalized to the sum of the frequency of occurrence of all mutations at the same position and is indicated by a blue color scale.” What is the rationale for the reference being the sum of the frequency of occurrence at that position? A mutation that is perfectly tolerated might get normalized to 50% if it and the wildtype are the only tolerated amino acid, or it might get normalized to 5% if all mutations are completely tolerated. Thus, there is the possibility of a 10-fold difference in the measure of tolerance for two mutations that are both completely neutral. Wouldn't it be better to normalize to the frequency of the synonymous mutation at that position, which serves as the control for neutral mutation creation efficiency?

We indeed divided the read counts of any specific edit by the total number of read counts that contained one of the twenty possible designed edits for that specific position. The rationale behind this normalization is that the efficiency of editing varies strongly throughout the targeted genes. This variability of editing is influenced by the efficiency and proximity of the targeted PAM site and is possibly also affected by other - hitherto unknown - factors. Although our original normalization method compensates for differences in editing efficiencies across the genes, we acknowledge that it can lead to a distorted interpretation of the data as explained by the reviewer. The alternative, normalizing to the read counts of the synonymous mutation, is a valid approach except for those positions where the synonymous edit is non-existent (M/W) or absent (10 synonymous mutations missing for the *fabZ* library, 17 for *lpxC* and 26 for *murA*).

However, because we have now performed a selection experiment interrogating each edit's fitness effect (explained in the response to comment 3), we were able to normalize the read counts of each edit at the end of the selection experiment to the read counts at the start of the experiment. This normalization method takes into account the differences in editing efficiency and has the added benefit of displaying fitness effects without being influenced by the composition of the input material used for transformation and construction of the saturation editing libraries (see response to comment 3). Additionally, the distorted data interpretation associated with our previous normalization method has been removed and missing synonymous mutations are not an issue. We therefore believe that this data representation is superior to what we have previously used and addresses the reviewer's comment.

5. Figure 2D-F. Whether a mutation is going to count as “successful” or not (as the authors are defining it) will depend on three things 1) the frequency at which that mutation was introduced into the genome, 2) the effects of that mutation on cell fitness, and 3) the number of generations of growth of the library after the mutation is introduced. The authors state they have tried to minimize #3, presumably so that they could observe any mutation that has a selection coefficient > -1 , even if the fitness cost was substantial. However, some growth is presumably necessary to enrich for tolerated mutations so that the mutation is not observed for cells that do not grow because the mutation is lethal. If this is the authors' intention, how do they know they have the right balance in the number of generations of growth? Also, the authors' approach does not seem to account for #1. For example, consider position A where mutations are made with great frequency and position B where mutations are made at low frequency. If the two positions were equally tolerant to mutation (but some mutations have deleterious fitness effects), using the authors' approach, position B would appear less tolerant. I would suggest that the authors are ignoring useful information in plotting Fig 2D-E: the

frequency at which the mutations are observed relative to the synonymous mutation. The authors view it binary (did the mutation appear or not). Perhaps instead (or in addition) the authors could sum up the frequencies relative to the synonymous mutation at that position (see my comment #4 above). That way if all mutations at a position reduced fitness by 90%, you would get a low value (instead of a 19 the way the authors are doing it).

The question of what the right balance in generations of growth is to reveal essential residues, is an interesting one. Unfortunately, we believe that there is not one single correct answer and that this is ultimately a semi-arbitrary decision. As may be expected, there is no threshold number of generations that clearly distinguishes between edits that abolish protein function and other edits that confer potentially significant fitness defects. Instead, there appears to be more of a continuum. No artificially defined cut-off will therefore be able to capture all biological nuance.

Nonetheless, a choice has to be made on which time point to take to analyze library composition. This was originally chosen to be the earliest time point where libraries could be harvested, i.e. right after construction on the Onyx platform. This optimized approach could indeed reliably determine the vast majority of important residues in LpxC and MurA. However, our selection experiment (explained in response to comment 3) has revealed that this is not universally true and that additional cycles of growth are necessary to reveal FabZ residues that are important for protein function and viability. This becomes obvious when looking at Figure 2 (also shown below). Whereas library composition is more or less stable for LpxC and MurA, multiple generations are required for FabZ library composition to stabilize. Because of these findings, and to remain consistent throughout our manuscript, we have now performed our analyses on FabZ, LpxC and MurA libraries that were grown for an additional 15 generations. Whereas the identified important residues for LpxC and MurA remain same, our approach is now able to discover more FabZ residues that are important for protein function. We want to thank the reviewers for their comments that have prompted us to look into the FabZ library in more detail, because we strongly believe that the new analyses presented in our revised manuscript represent a significant improvement to our work.

Figure 2: The stability of library composition across growth cycles varies for different proteins. A-C) The cumulative frequency distribution of successful amino acid substitutions, i.e. edits that could be detected by sequencing, is shown for the FabZ (A), LpxC (B) and MurA (C) libraries. Different curves represent the library

composition at different points of sequencing. In case multiple replicates were sequenced (for samples “5” and “15”), the first replicate is shown. D-F) This figure shows the dropout fraction, i.e. the fraction of edits detected by fewer than 10 reads, in function of the number of generations the FabZ (D), LpxC (E) or MurA (F) libraries were grown. As a reference, the dropout fraction at the start of the selection experiment is indicated by a dotted line. “Original” refers to the libraries immediately after construction. Generations 0, 5 and 15 refer to the number of generations libraries were grown as part of the selection experiment. As expected, cycles of growth will lead to an increased loss of amino acid substitutions, resulting in less steep cumulative distribution curves and larger dropout fractions. AA, amino acid.

The second part of this comment states that we should normalize edit frequencies to the synonymous mutation to get information on the fitness effects of each edit. Whereas we value this comment and acknowledge that it is better suited to reveal potential fitness effects than our originally chosen normalization method, we are hesitant to apply it because we believe that read counts of individual edits are not directly tied to their fitness but will also be influenced by other factors such as the composition of the input pool of editing cassettes used for transformation. Additionally, with the data from the selection experiment at hand, we now have a superior data set to characterize fitness effects of introduced mutations. These fitness effects are presented in Results paragraph “Competition within saturation editing libraries reveals the fitness effect of each edit” (starting at line 148), Tables S3 and S4 and Figures 2, 4, S2, S3 and S4.

6. The authors note surprising observations of mutations that are tolerated that seem to contradict previous studies on these proteins. In addition, the authors found FabZ to be remarkable tolerant to mutation given its essential nature. One possibility the authors do not consider is the frequency of mistranslation. Codons in an mRNA message are prone to mistranslation errors in ways that depend on the mutant codon, the neighboring codons, and the bacterial strain being used. Perhaps the level of active protein necessary for the authors to observe a mutation and mark it as successful is very low (say 1% or 0.1% of the wildtype level). This can happen if fitness is greatly buffered to changes in total cellular protein activity, which has been observed for some genes. Thus, mistranslation to incorporate the wildtype (or another tolerated) amino acid at a mutated position would not need to be very efficient for that mutation to appear tolerated even though a protein with that mutation is inactive, for example. I’m not suggesting this possibility seriously impacts that authors’ findings, after all the mutation is still tolerated regardless of the mechanism. However, I think the authors should discuss whether mistranslation might impact or explain some of their results, especially the apparent contradictions with previous studies (e.g. in vivo vs. in vitro differences).

We thank the reviewer for this comment since we had not previously considered this particular mechanism of tolerance. We have now added this possibility to our revised manuscript at line 509.

Minor

1. Line 130. Extra period

Thank you for drawing our attention to this typo. We have fixed it in our revised manuscript.

REVIEWERS' COMMENTS

Reviewer #1 (Remarks to the Author):

The authors have responded to the points in my original review. In particular, they have extended the selection to multiple generations to obtain fitness measurements, giving increased sensitivity to small effects and quantitating fitness. The manuscript is substantially improved.

Reviewer #2 (Remarks to the Author):

The authors revised the manuscript and added sufficient information to the revised manuscript. I am satisfied with the manuscript and I believe that it is good to publish in Nature Communications.

Reviewer #3 (Remarks to the Author):

The authors have satisfactorily addressed my comments.